# PAX3-FOXO1 uses its activation domain to recruit CBP/P300 and shape RNA Pol2 cluster distribution

Yaw Asante [1,7], Katharina Benischke[2,7], Issra Osman [3,7], Quy A. Ngo[2], Jakob Wurth [2], Dominik Laubscher[2], Hyunmin Kim [3], Bhavatharini Udhayakumar[1], Md Imdadul H. Khan [3], Diana H. Chin [3], Jadon Porch[3], Maharshi Chakraborty[4], Richard Sallari[4], Olivier Delattre [5], Sakina Zaidi [5], Sarah Morice[6], Didier Surdez [6], Sara G. Danielli[2], Beat W. Schäfer [2,8] ✉, Berkley E. Gryder [3,8] ✉ & Marco Wachtel [2,8] ✉

Activation of oncogenic gene expression from long-range enhancers is initiated by the assembly of DNA-binding transcription factors (TF), leading to recruitment of co-activators such as CBP/p300 to modify the local genomic context and facilitate RNA-Polymerase 2 (Pol2) binding. Yet, most TF-to-coactivator recruitment relationships remain unmapped. Here, studying the oncogenic fusion TF PAX3-FOXO1 (P3F) from alveolar rhabdomyosarcoma (aRMS), we show that a single cysteine in the activation domain (AD) of P3F is important for a small alpha helical coil that recruits CBP/p300 to chromatin. P3F driven transcription requires both this single cysteine and CBP/p300. Mutants of the cysteine reduce aRMS cell proliferation and induce cellular differentiation. Furthermore, we discover a profound dependence on CBP/p300 for clustering of Pol2 loops that connect P3F to its target genes. In the absence of CBP/p300, Pol2 long range enhancer loops collapse, Pol2 accumulates in CpG islands and fails to exit the gene body. These results reveal a potential novel axis for therapeutic interference with P3F in aRMS and clarify the molecular relationship of P3F and CBP/p300 in sustaining active Pol2 clusters essential for oncogenic transcription.

Rhabdomyosarcoma (RMS) is an aggressive cancer that forms neoplasm of mesenchymal origin in soft-tissues and hollow organs, primarily in children and adolescents[1,2]. The majority of alveolar RMS (aRMS) is defined by a unique fusion TF that alters the transcriptional program (fusion-positive RMS; FP-RMS). This protein is a result of a chromosomal translocation event, t(2;13)(q35;q14), that produces an in-frame fusion of the transcription factors PAX3 and FOXO1 and represents the most common fusion event in FP-RMS, the most devastating RMS subtype. PAX3-FOXO1 (P3F) contains the N-terminal DNA-binding domains of PAX3, the paired box and the homeobox, fused to the C-terminal domain of FOXO1 mediating transactivation. While FP-RMS cells are addicted to the fusion protein, as a

[1]Department of Nutrition, Case Western Reserve University, Cleveland, OH, USA. [2]University Children's Hospital, Children's Research Center and Department of Oncology, Steinwiesstrasse 75, CH-8032 Zürich, Switzerland. [3]Department of Genetics and Genome Sciences, Case Western Reserve University, Cleveland, OH, USA. [4]Axiotl Inc, Cleveland, OH, USA. [5]INSERM U830, Diversity and Plasticity of Childhood Tumors Lab, PSL Research University, SIREDO Oncology Center, Institut Curie Research Center, Paris, France. [6]Balgrist University Hospital, Faculty of Medicine, University of Zurich (UZH), Zurich, Switzerland. [7]These authors contributed equally: Yaw Asante, Katharina Benischke, Issra Osman. [8]These authors jointly supervised this work: Beat W. Schäfer, Berkley E. Gryder, Marco Wachtel. ✉e-mail: beat.schaefer@kispi.uzh.ch; berkley.gryder@case.edu; marco.wachtel@kispi.uzh.ch

transcription factor, P3F is a notoriously challenging drug target. Indirect targeting strategies might be applied as an alternative to treat FP-RMS tumors[3].

Epigenetic regulators with druggable domains are of special interest in this context. Different epigenetic co-regulators of P3F have been identified in the last couple of years including BRD4 and CHD4 (ref. 4–6). As a result of their functional interplay, chromatin is opened at specific sites and allows other cofactors, including *MYOD1*, *MYCN*, and *MYOG*, to bind, establishing myogenic super enhancers capable of remodeling chromatin topography, miswiring muscle development programs, and activating tumorigenesis[4,7]. While the interaction of P3F with these epigenetic factors is DNA dependent, the exact mechanism of action and nature of all proteins involved in the complex assembled around P3F is still largely unknown. Importantly, a better understanding of the mode of action of P3F at the molecular level, including detailed structure-function relationship to identify functionally important domains, may provide new strategies to drugging P3F for therapeutic benefit.

In this study, we used CRISPR based domain screening in combination with mutational analysis to characterize the activation domain

(AD) of P3F. We discovered how a single cysteine residue in the AD forms a critical hook for recruiting CBP/p300, leading to the expression of P3F target genes. Upon mutation of this cysteine, FP-RMS cells, normally arrested at an early-stage of the myogenic differentiation process, become more mature, taking on a myocyte-like morphology. We also find that CBP/p300 activity enables topographical connectivity of Pol2 between P3F-bound enhancers and target promoters. Degradation of CBP/p300 by dCBP1 treatment selectively halts P3F's gene expression program in different PDX-derived primary FP-RMS models, suggesting a potential therapeutic relevance.

## Results

### A novel functional unit is present inside the PAX3-FOXO1 activation domain

Given that P3F is essential to FP-RMS growth, we employed a tiling CRISPR/Cas9 mutagenesis screen to map functionally relevant protein domains[8]. RH4 cells expressing Cas9 were transduced with one of two vector constructs: one with a control sgRNA targeting the AAVS1 locus and expressing BFP, and the other with a sgRNA targeting P3F and expressing RFP (Fig. 1a). A total of 77 P3F-targeting sgRNAs were

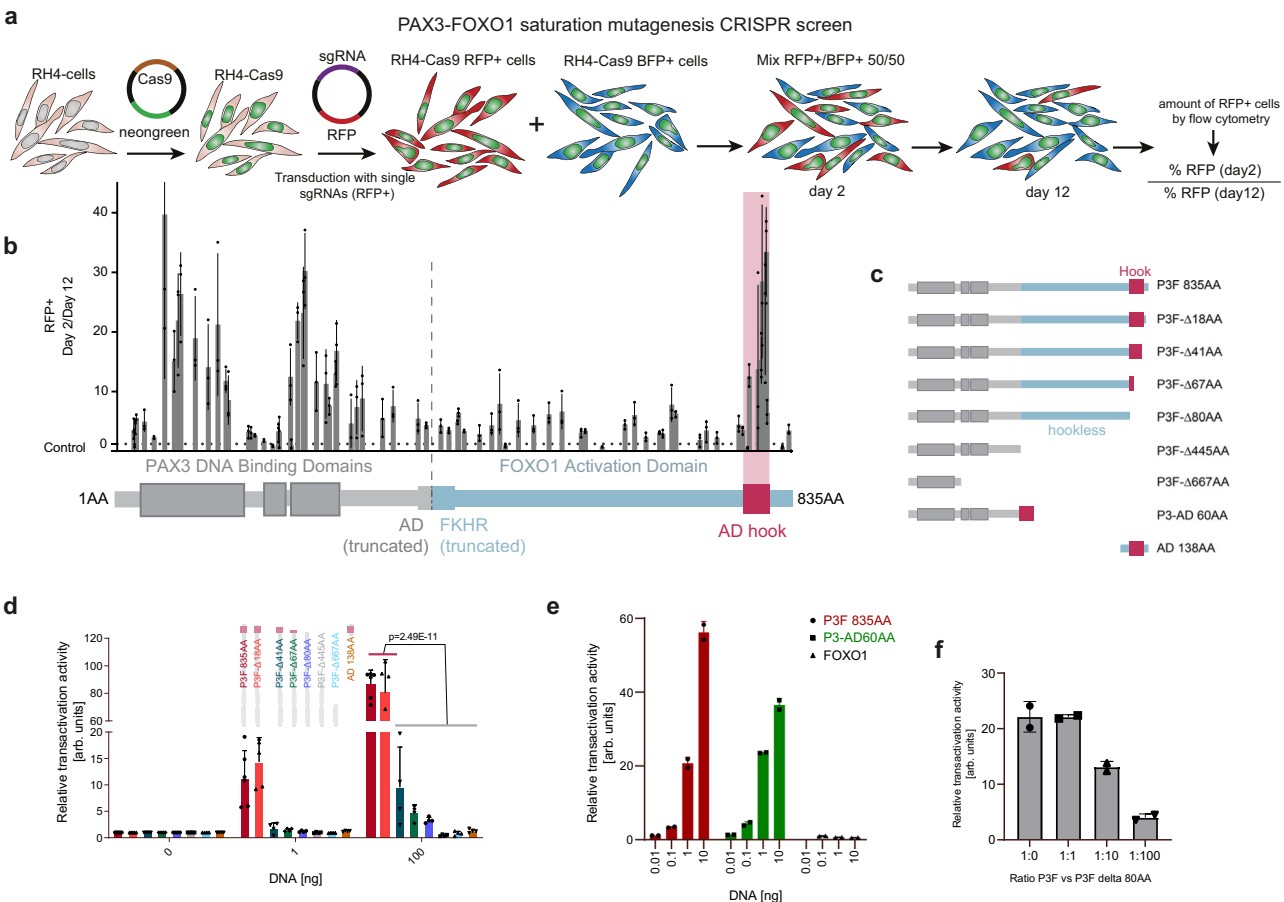

**Fig. 1 | CRISPR/Cas9-based domain screening of P3F reveals a novel functionally important C-terminal domain. a** Scheme depicting the CRISPR/Cas9-based domain screen approach. RH4 cells stably expressing Cas9 (RH4-Cas9) were transduced with either a vector driving expression of sgRNAs directed against P3F and RFP or a control sgRNA directed against the AAVS1 region together with BFP. Two days after transduction RFP+ cells were mixed 1:1 with BFP+ cells. Percentage of RFP+ cells was determined at day 2 and 12 by flow cytometry. **b** Ratio of RFP+ cells on day 2 (D2) and day 12 (D12). Relative number of RFP+ cells was measured by flow cytometry. The horizontal dashed line indicates the mean of all controls (ratio 1.22). Plotted are mean and standard deviation for each sgRNA (*n* = 3 independent experiments). P3F domains (PAX3 in grey, FOXO1 in blue) are depicted schematically at the bottom, with the vertical dashed line indicating the breakpoint of the

fusion. AD, activation domain; FKHR, forkhead domain. **c** Scheme depicting the truncated versions of P3F used for reporter assays. **d** Luciferase reporter assays measured 48 h after transfection of HEK 293 T cells with indicated P3F constructs. Depicted are mean and standard deviation for each construct normalized to an internal transfection control (Renilla luciferase) (*n* = 6 (P3F 835 AA), *n* = 5 (P3F-Δ18AA) and *n* = 4 (rest) independent experiments; two-way Anova, Tukey's multiple comparisons test). **e** Luciferase assay performed as described under d using full-length P3F, a 60 AA fragment containing the AD fused to the PAX3 part of P3F (P3-AD 60AA) or wildtype FOXO1 (*n* = 2 independent experiments). **f** Luciferase assay performed as described under d with full-length P3F in combination with the P3F-Δ80AA deletion mutant in indicated ratios (*n* = 2 independent experiments). Source data are provided as a Source Data file.

synthesized to cover the whole sequence of P3F. As controls, we used 9 sgRNAs targeting either exon 8 of PAX3 or exon 1 of FOXO1, which are not included in the fusion protein (Supplementary Data S1). Two days after transduction with either vector, cells were mixed 1:1, grown for 12 days, and the remaining population of RFP+ cells was determined by flow cytometry. The sgRNAs that led to a depletion of RFP+ relative to BFP+ cells allowed scoring of functional regions, whereby regions of high sensitivity form clear hotspots with multiple high-scoring sgRNAs (Fig. 1b). This highlighted not only the essentiality of both PAX3-DNA binding domains, but also identified a small but critical region towards the C-terminal end of FOXO1, suggesting that this may be the heart of transcriptional activation. Interestingly, although the majority of the FOXO1 part is intrinsically disordered (Supplementary Fig. S1a) and may play a role in transcriptional condensate formation, the functionally sensitive component of the FOXO1 domain was a region predicted to be ordered as an alpha helix (Supplementary Fig. S1b).

To further validate the relevance of the small C-terminal domain for P3F function, we built several constructs of truncated P3F (Fig. 1c, Supplementary Fig. S1c) and transfected them into HEK 293T cells together with a P3F-responsive luciferase reporter based on the P3F binding site of its target gene *ASS1* (Supplementary Fig. S1d)[9]. Results from this reporter assay showed that small deletions C-terminal to the identified small AD (P3F -Δ18AA) did not reduce the transactivation potency of P3F. In contrast, truncations (P3F-Δ41AA, P3F-Δ67AA) or complete deletion (P3F-Δ80AA) of the small AD led to a near complete loss of P3F-induced luciferase expression, similar to depletions of the whole FOXO1 part (P3F-Δ445AA) or larger parts of the protein (P3F-Δ667AA) (Fig. 1d). A complementary result was seen in a reporter assay with a negative control reporter containing a mutant P3F-binding site, with only the smallest truncation matching full-length transactivation (Supplementary Fig. S1e). Importantly, with exception of P3F-Δ667AA, all deletion mutants are predominantly (from full-length P3F down to P3F-Δ445AA) or partially (AD 138AA) localized in the nucleus (Supplementary Fig. S1f). As expected, expression of the C-terminal AD alone (AD 138AA) was also not sufficient to promote luciferase expression (Fig. 1d). However, when fused to the PAX3 part of P3F, a 60AA long fragment spanning the AD had a transactivation potency similar to full-length P3F (Fig. 1e), confirming that this small part suffices for induction of transactivation and demonstrating that the whole disordered part of FOXO1 is not relevant, at least in this assay. Wildtype FOXO1 did not transactivate this reporter, highlighting the importance of specific DNA binding (Fig. 1e). Furthermore, addition of different amounts of the inactive P3F-Δ80AA deletion mutant to full-length P3F reduced transactivation in a dose-dependent manner (Fig. 1f), as expected in case of competition for DNA binding. Hence, these data demonstrate that a 40 AA long C-terminal AD is required for full transcriptional activity of P3F.

## A single cysteine in the FOXO1 domain is necessary for transcriptional activity of PAX3-FOXO1

We next aimed to characterize the C-terminal AD in more detail. First, we wondered whether it would be possible to identify individual AAs in this domain that are required for P3F activity. Indeed, we identified a cysteine (in P3F, C793; in FOXO1, C612), which is highly conserved among all FOXO family members as well as among FOXO1 from different species (Fig. 2a and Supplementary Fig. S2a) and which is important for the function of wild-type FOXO proteins[10]. Therefore, we mutated C793 as well as several amino acids N- and C-terminal to it, including D792, D794, M795, E796 and S797 by site directed mutagenesis and assessed the mutant's transactivation potency by luciferase assays as described before (Fig. 2b). Interestingly, mutation of C793 to serine decreased luciferase expression by more than 50 percent compared to wild-type (wt) P3F ($p = 0.0007$) (Fig. 2b and Supplementary Fig. S2b). A reduction by 40% was detected upon mutation

of the neighboring D794, whereas mutations of all other amino acids had only little or no influence on reporter gene transactivation. To determine whether this was not a general effect of cysteine mutation, we also mutated C765 to a serine. However, this mutation had no influence on the transactivation potency of P3F (Fig. 2b). Consequently, C793 plays a central role for the activity of P3F.

To verify that C793 also plays an important role for P3F function in FP-RMS cells, we developed a doxycycline(dox)-inducible shRNA to specifically silence endogenous P3F expression (shP3F) (Supplementary Fig. S2c). RH4 cells containing the shRNA construct were then rescued with either an empty expression vector (shP3F/ev) as control, wt P3F (shP3F/P3F wt) or C793S mutant P3F (shP3F/P3F C793S), with all these being shRNA resistant by containing silent mutations in the binding site. Interestingly, presence of P3F C793S induced morphological changes in the cells even before the endogenous P3F was silenced (shP3F-dox/P3F C793S), with cells adopting an elongated shape reminiscent of myogenic differentiation (Fig. 2c). Also, proliferation was significantly impaired compared to cells transduced with empty vector (shP3F-dox/ev) or wt P3F (shP3F-dox/P3F wt) (Fig. 2d). Strikingly, after shRNA induction with dox for five days, a reduction of cell numbers was observed when cells expressed the mutant P3F (shP3F+dox/P3F C793S), similar to single knockdown of P3F, and to much lower levels when compared to cells transduced with wt P3F (shP3F+dox/P3F wt). Analysis of gene expression after 48 h of dox-treatment showed that knockdown of endogenous P3F alone (shP3F +dox/ev) leads to significant upregulation of several myogenic differentiation markers, which was completely rescued by expression of wt P3F (shP3F+dox/P3F wt), but not by the C793S mutant (shP3F+dox/P3F C793S) (Fig. 2e). In this, two of the three differentiation markers (MYL1, TNNC2) were already significantly upregulated in the absence of dox, with no further upregulation after addition of dox. Similar findings were observed for TNNC2 in an additional FP-RMS cell line (KFR) (Supplementary Fig. S2d). Interestingly, disorder prediction suggests that the C793S mutation potentially weakens secondary structures in this region (Fig. 2f). Finally, these findings were further substantiated by site-directed mutagenesis of C793 in endogenous P3F in RH4 cells using an adenine-deaminase based CRISPR base editing approach[11]. Using this strategy, the cysteine at position 793 was mutated into arginine with very high efficiency (Fig. 2g). Expression of P3F protein remained intact and detectable in the nucleus as transcriptional hubs as determined by super-resolution microscopic analysis of immuno-fluorescence staining using a P3F-specific antibody[12] (Supplementary Fig. S3a, b). Importantly, cells with C793R mutant P3F heavily differentiated and upregulated markers of terminal muscle differentiation, including *MYH3* and *TNNC2*, while expression of endogenous P3F target gene *ASS1* was drastically reduced (Fig. 2h, i). In a competition assay similar to the one used for the screen to evaluate effects on cell survival, mutation of C793 led to an 8-fold depletion of affected cells compared to control cells, confirming the relevance of intact C793 in an endogenous context (Supplementary Fig. S3c). Interestingly, both silencing and mutation of P3F lead to an upregulation of wildtype FOXO1 expression (Supplementary Figs. S2c and 3a). To evaluate whether FOXO1 is involved in some of the downstream effects after P3F inactivation, we generated a FOXO1 knock-out variant of RH4 cells by mutation of the splice site of exon 1 using base editing (Supplementary Figure S4a, b). Interestingly, upregulation of muscle differentiation markers was strongly reduced in these cells upon mutation of C793 of P3F compared to parental cells, while effects on P3F target genes were less affected (Supplementary Fig. S4c). This suggests that part of the physiological effect downstream of inactivation of P3F is indeed due to FOXO1 upregulation. Taken together, these results demonstrate that the transactivation potency of P3F is highly dependent on a single cysteine in the small C-terminal AD and that interference with this amino acid affects proliferation and induces differentiation of FP-RMS cells.

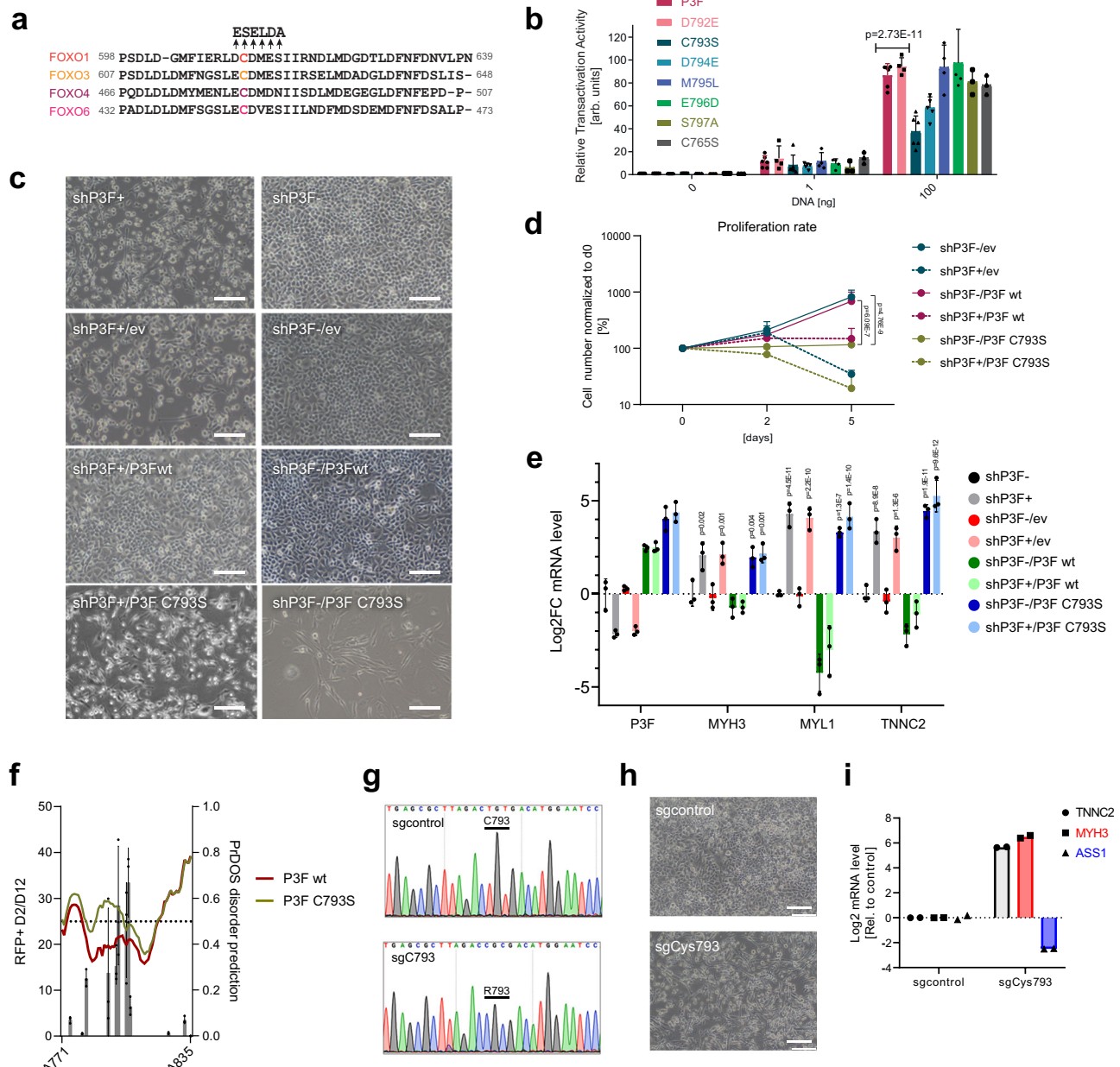

**Fig. 2 | Mutation of C793 reduces transcriptional activity of P3F and induces differentiation of FP-RMS cells. a** Alignment of amino acid sequences of FOXO family members around C612 of FOXO1 (C793 of P3F) (based on UniProt). Arrows indicate amino acids that were mutated in P3F for functional tests. **b** Luciferase reporter assay performed with HEK293T cells 48 h after transfection with either wildtype P3F or P3F containing indicated mutations. Plotted are mean ± SD of each construct normalized to an internal transfection control (Renilla luciferase) ($n = 6$ (P3F, C793S), $n = 5$ (D794), $n = 4$ (D792, M795, E796) and $n = 3$ (S795, C765) independent experiments, two-way Anova, Tukey's multiple comparisons test). **c** Morphology of RH4 cells before (right panels) and after (left panels) silencing of endogenous P3F (shP3F) using a dox-inducible shRNA and rescued with either empty vector (ev), wild type P3F (P3Fwt) or C793S mutant P3F (C793S), respectively. Scale bar, 100 µm. **d** Proliferation curve of cells described in c, as determined by cell counting ($n = 3$ independent experiments; means ± SD, two-way Anova, Tukey's multiple comparisons test). **e** mRNA levels of indicated genes in cells

described in c 48 h after silencing of P3F. Data is normalized to shP3F d0 control. Plotted are means ± SD ($n = 3$ independent experiments, two-way Anova, Tukey's multiple comparisons test, comparison to shP3F). **f** Disorder prediction of the C-terminal region (amino acid 777-831) of wildtype (red) and C793S mutant (green) P3F calculated with PrDOS (www.predictprotein.org). Results of the domain screen from b are overlaid as black bars ($n = 3$ independent experiments, plotted as means ± SD). **g** Sanger sequencing chromatograms depicting the region around C793 in endogenous P3F before and after base editing in Rh4-ABE8e cells. An amplicon covering C793 was amplified with cDNA from cells transduced with a control sgRNA (upper panel) or an sgRNA recruiting the ABE8e base editor to C793 (lower panel). **h** Phase contrast pictures of the same cells as described in g, 3 days after transduction. Scale bar, 400 µm. **i** mRNA levels of indicated genes measured in cells as described in g 5 days after transduction ($n = 2$ independent experiments). Source data are provided as a Source Data file.

## C793 is necessary to establish the PAX3-FOXO1 target gene signature

To interrogate the relevance of C793 for expression of the P3F target gene signature in FP-RMS cells, we performed RNA-seq analysis in Rh4

cells containing the rescue system described above. We found that cells expressing empty vector (shP3F + 24 h/ev and shP3F + 48 h/ev) or the C793S mutant (shP3F + 24 h/C793S and shP3F + 48 h/C793S) showed similar clustering along their principal components (Fig. 3a),

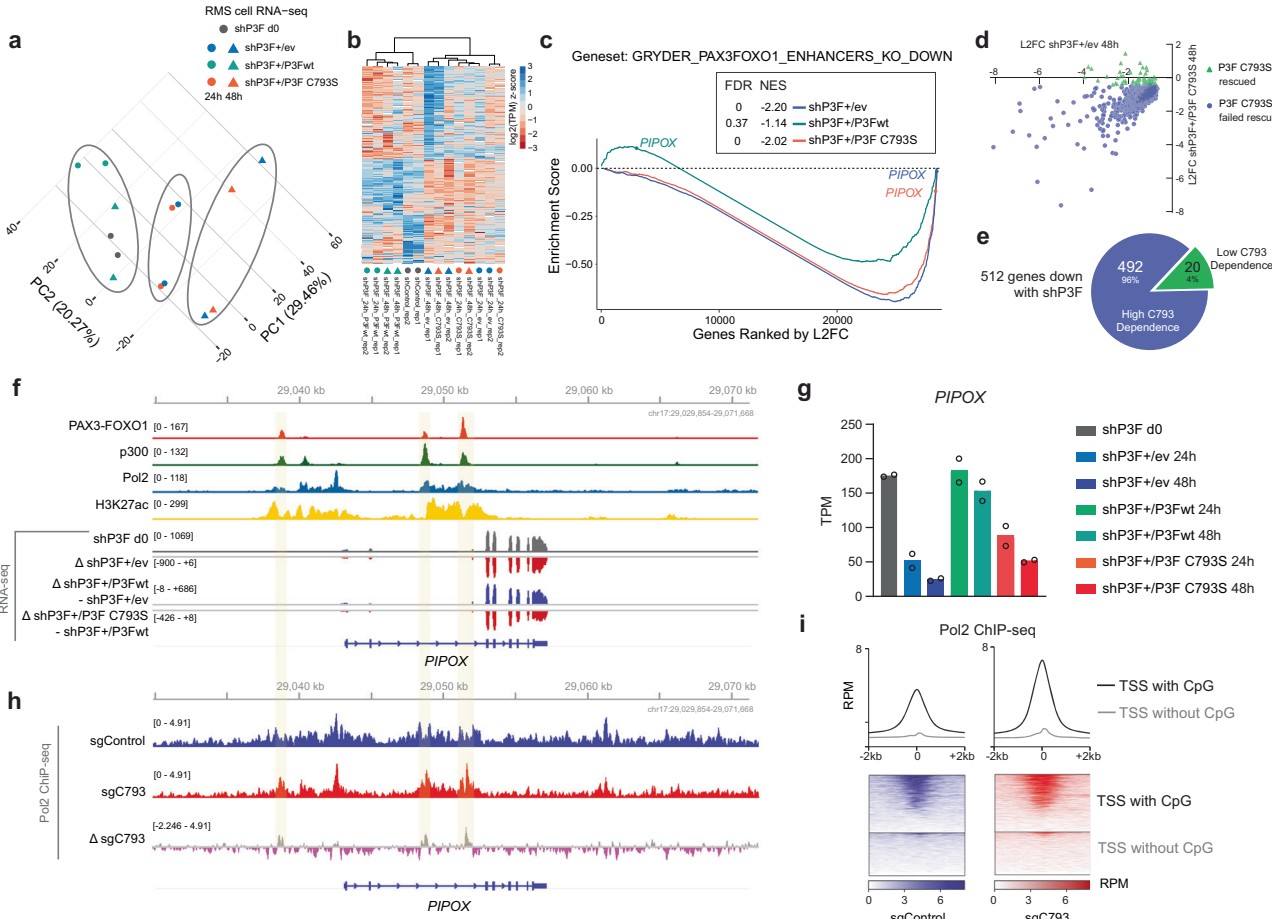

**Fig. 3 | Mutation of C793 leads to loss of expression of most of the P3F target genes.** RNA-seq analysis performed with RH4 cells after silencing of endogenous P3F with a Dox-inducible shRNA and rescue with either empty vector (ev), wild type P3F (P3Fwt) or P3F carrying a C793S mutation (C793S), respectively, for 24 h and 48 h (*n* = 2). Non-dox treated cells containing no rescue construct were used as baseline control (shP3F d0). **a** Principal component analysis (PCA) plot displaying 14 samples. **b** Unsupervised hierarchical clustering analysis using all genes up- or downregulated after silencing of endogenous P3F and rescued with wildtype P3F (*n* = 854) in RH4 cells (FDR < 0.05). **c** GSEA analysis performed using shP3F + /ev 48 h, shP3F + /P3Fwt 48 h, and shP3F + /P3F C793S 48 h data. Pre-ranked datasets were prepared using log-transformed data normalized with shP3F d0 baseline control construct data. The gene set was derived from a list of P3F target genes recurrent in RMS tumors and cell lines that fall within TADs containing P3F-bound enhancers. FDR, false discovery rate; NES, normalized enrichment score. **d** Changes in expression of P3F target genes (*n* = 512) after silencing of endogenous P3F in presence and absence of C793S mutant P3F. **e** Pie graph depicting genes with high and low C793 dependence among the 512 most downregulated genes after silencing of endogenous P3F at 48 h in RH4 cells. The majority of these genes have a high C793 dependence (*n* = 492). **f** ChIP-seq and RNA-seq data at *PIPOX* visualized in IGV. **g** TPM values of *PIPOX* across all construct treatments. Error bars represent the standard deviation across two biological replicates. **h** Pol2 ChIP-seq data for RH4 cells after base editing with sgcontrol (wildtype P3F) versus C793R mutant P3F at *PIPOX* visualized in IGV. Overlap between P3F sites and delta tracks highlighted in yellow. **i** Heatmap of Pol2 ChIP-seq shows that TSS with CpG islands have an increase in signal after P3F C793R mutation which is greater than that at the TSS without CpG islands. Source data are provided as a Source Data file.

while samples rescued with wt P3F (shP3F + 24 h/P3Fwt and shP3F + 48 h/P3Fwt) clustered together with the untreated control (shP3F d0). Groupwise comparisons identified 854 genes which are either up- or downregulated with an FDR of less than 0.05 after silencing of endogenous P3F compared to control (Supplementary Data S2). 512 of these genes were downregulated, while 342 were upregulated upon P3F silencing. The downregulated genes represent P3F target genes, while among the upregulated genes many are induced by the myogenic differentiation process initiated after loss of P3F. Hierarchical clustering analysis shows that at the early time point (24 h dox) most of the P3F target genes are already silenced, while upregulation of genes is delayed (Fig. 3b). This analysis also reconfirms the clustering of control samples (shP3Fd0) together with P3F wt cells (shP3F+dox/P3Fwt), indicating rescue of the P3F target gene signature by overexpression of wtP3F, while loss of the P3F target gene signature was observed in presence of the C793S mutant. Gene set enrichment analysis (GSEA) of target genes in the same topologically associated domain (TAD) with P3F (ref. 4) revealed that control and P3F/C793S cells downregulated a

similar number of genes compared to those rescued with the wt P3F (Fig. 3c). Also, the top 20 gene ontology terms found enriched in downregulated genes after 48 h dox treatment of cells transfected with empty vector (shP3F + dox48h/ev) and mutant P3F (shP3F + dox48h/C793S) were highly similar (Supplementary Fig. S5a). Correlation analysis of transcript levels of genes affected by silencing of endogenous P3F in presence and absence of C793S mutant P3F revealed that gene expression changes are highly similar (Fig. 3d). In total, 492/512 (96%) P3F dependent genes are not re-activated by the P3F C793S mutant (Fig. 3e). To illustrate an example of how the C793S mutation affects known P3F target genes, we investigated *PIPOX*, a gene that is known to be regulated by a P3F-induced super-enhancer located in intron 2 (ref. 4). Published ChIP-seq data from FP-RMS cells[13] confirmed that p300, RNA Pol2, P3F, and H3K27ac are all highly enriched at the *PIPOX* locus (Fig. 3f), highlighting its importance as a marker for P3F activity. RNA-seq delta tracks as well as TPM data both demonstrate that the empty vector and the C793S P3F mutant failed to rescue *PIPOX* transcription after endogenous P3F silencing, unlike wt P3F (Fig. 3f, g).

Similar patterns were seen for other PAX3-FOXO1 target genes such as *ASS1*, *CDH4*, and others (Supplementary Fig. S5b). Then, we compared RNA Pol2 ChIP-seq between RH4 cells containing either wild-type or C793R mutant P3F. While we saw that the C793R mutation induced a small, genome-wide increase in Pol2 binding at P3F binding sites (Fig. 3h and Supplementary S5c), the most striking finding was a substantial increase of Pol2 near TSS which contain overlapping CpG islands (Fig. 3i).

### C793 recruits CBP/p300 to co-occupy enhancer sites

To study the mechanisms by which C793 might affect transcriptional activity of P3F, we next analyzed the P3F interactome. To identify protein-protein interactions of wildtype P3F versus C793S mutant P3F, we performed a BioID experiment in HEK293T cells which were transiently transfected with a BirA-Flag/P3F or a BirA-Flag/P3F C793S construct in presence of biotin (Fig. 4a, left). 24 h after transfection, biotin-labelled proteins were isolated from cell lysates using streptavidin affinity columns and identified by mass spectrometry (Supplementary Fig. S6a and Supplementary Data S3). We identified 12 proteins for which spectral counts were reduced by at least 50 percent in C793S mutant compared to wt P3F (Fig. 4a). Six of these 12 proteins were found to be essential for FP-RMS cell survival in published genome-wide CRISPR knockout screen data (depmap.org, Supplementary Fig. S6b). Among these, p300 has previously been shown to act as co-factor for wildtype FOXO transcription factors[14] and was therefore further investigated. BioID followed by Western blot confirmed the reduced ability of C793 mutant P3F-BirA to biotinylate p300 compared to the wildtype P3F-BirA. In contrast, auto-biotinylation was unaffected, while biotinylation of HDAC2, which as a member of the NuRD complex is detected at genomic sites in close proximity to P3F binding sites[5,6] was much less affected by the mutation compared to p300 (Fig. 4b). Since we were not able to confirm the interaction between P3F and p300 by co-immunoprecipitation, we performed a previously described recruitment assay to validate this interaction. This assay is based on U2OS cells bearing 50,000 repeats of the Lac operon integrated into one single site on chromosome 1[15]. Fusion proteins containing the LacI DNA binding domain and the cyan fluorescent protein (CFP) are highly enriched at this chromosomal site and can be detected as bright fluorescent spots, while recruited interactors are detected by immunofluorescence at the same site. We transfected these cells with different LacI-CFP-FOXO1 fusion protein constructs, followed by immunofluorescent detection of p300 (Fig. 4c). This assay revealed that the FOXO1 part of P3F is able to recruit p300, whereas a 90 percent reduction was observed when the small C-terminal AD was deleted (Figs. 4d and 4e). Interestingly, recruitment was unaffected when the complete intrinsically disordered domain of FOXO1 was deleted, but the small AD present (Fig. 4e). Furthermore, the C793S mutant FOXO1 domain was about 30 percent less effective than the wt FOXO1 in recruiting p300. To identify the p300 domains involved in this interaction, we also generated fusion proteins between mCherry and different p300 domains known to be involved in protein-protein interaction (Supplementary Fig. S6c). This approach demonstrated that among individual p300 domains only the KIX domain is weakly recruited by the FOXO1 AD (Supplementary Fig. S6c). Combinations of different p300 domains, however, including ZZ-TAZ2-IBID and the TAZ1-KIX-ZZ-TAZ2-IBID fusions were recruited more efficiently (Supplementary Fig. S6c).

Given that P3F and p300 can bind to each other, we hypothesized that this would result in enhanced co-occupancy at P3F-sites genome wide. We compared the average ChIP-seq signal of p300 peaks at sites co-occupied by P3F to p300 signals at sites where P3F is absent. Strikingly, p300 levels are 2-3 times higher at P3F sites than at control sites, in accordance with higher H3K27Ac levels, while no difference in DNA accessibility was detected (Fig. 4f). This supports the hypothesis that P3F assists in directing p300 to result in target gene activation.

Next, we aimed to address the contribution of p300 to P3F target gene expression. Since p300 is highly homologous to the cyclic AMP response element-binding protein (CBP), and both proteins have the common role of co-regulators of FOXO proteins, we generated single and combined knockouts of p300 and CBP. RH4-Cas9 cells were transduced with sgRNAs targeting the histone acetyltransferase (HAT) domains of p300 and CBP individually or in combination as well as with controls (Supplementary Fig. S7a). Three days after transduction, RNA was isolated for gene expression analysis. Detection of different P3F target genes including *ALK*, *ASS1* and *FGFR4* as well as myogenic markers including *MYL1* and *TNNC2* by qRT-PCR showed that only the combined knockout of p300 and CBP has a strong effect on expression of these genes (Supplementary Fig. S7b). We therefore performed gene expression profiling using RNA-seq with the combined knockout samples. PCA and unsupervised hierarchical clustering with this data revealed striking differences between knockout and control samples (Fig. 4g and Supplementary Fig. S7c). GSEA results revealed that a double knockout of p300 and CBP downregulated known P3F target genes (Fig. 4h)[4].

Overall, these data show that the majority of the P3F target genes are co-regulated by CBP/p300. The small C-terminal AD of FOXO1 is essential for CBP/p300 recruitment and C793 is involved in this interaction. The AD domain appears to have a promiscuous interaction behavior with different CBP/p300 domains as described for other proteins, including p53 (ref. 16), with the individual interactions contributing additively to the overall interaction strength. Surprisingly, the intrinsically disordered part of FOXO1 appears to be dispensable for the recruitment of CBP/p300, indicating that this part might play a negligible role in this context.

### PAX3-FOXO1 requires CBP/p300 to maintain vital core-regulatory gene expression

Given that CBP and p300 are druggable enzymes with available inhibitors, we wondered whether such drugs would be of use to interfere with P3F in FP-RMS cells. We performed a six-day treatment of RH4, KFR, RMS13, and myoblast cells with increasing concentrations of A485 or A486, its negative control. We found that inhibition of CBP/p300 via A485 downregulated P3F target genes (Supplementary Fig. S7d) and caused more cell death in RH4, KFR, and RMS13 cell lines compared to A486; myoblasts showed a minimal difference in cell death between the two drug treatments (Figs. 5a, b). Selective potency caused us to wonder if HATi had selective transcriptomic effects, perhaps rooted in the disproportional p300 deposits in the genome described above. We used p300 ChIP-seq data from RH4 cells to rank genes by p300 quantity in their enhancers. We found that important core regulatory circuitry genes like *MYOD1*, *MYCN*, and *SOX8* were amongst the genes with the highest p300 signal (Fig. 5c). Analysis of changes in gene expression after treatment with A485 revealed a correlation between p300 load and expression changes after treatment. Accordingly, *MYOD1*, *MYCN*, and *SOX8* were among the most downregulated genes, demonstrating log2 fold changes ranging between −2 and −6 (Fig. 5d). In addition to A485, we also evaluated the p300/CBP degrader molecule dCBP1 (ref. 17). dCBP1 was even more effective in halting the growth of RH5 cells (Fig. 5e, top) and more potently perturbed the transcriptome already after 6 h (Fig. 5e, bottom). dCBP1 efficiently downregulated P3F target genes and the core regulatory transcription factors (CR TFs) of FP-RMS. At the same time, dCBP1 and A485 both only minimally downregulated housekeeping genes (Fig. 5f). TPM values for two of these CRTFs, *MYOD1* and *MYCN*, were lower after CBP/p300 degradation compared to CBP/p300 inhibition in both RH4 and RH5 cells (Fig. 5g). HAT inhibition or p300/CBP degradation downregulated FP-RMS CR TFs strongly and consistently across both cell lines (Fig. 5h, top panel) and also shut down known P3F target genes (Fig. 5h, bottom panel) with degradation again showing more pronounced effects. This suggests that high enrichment of p300/CBP

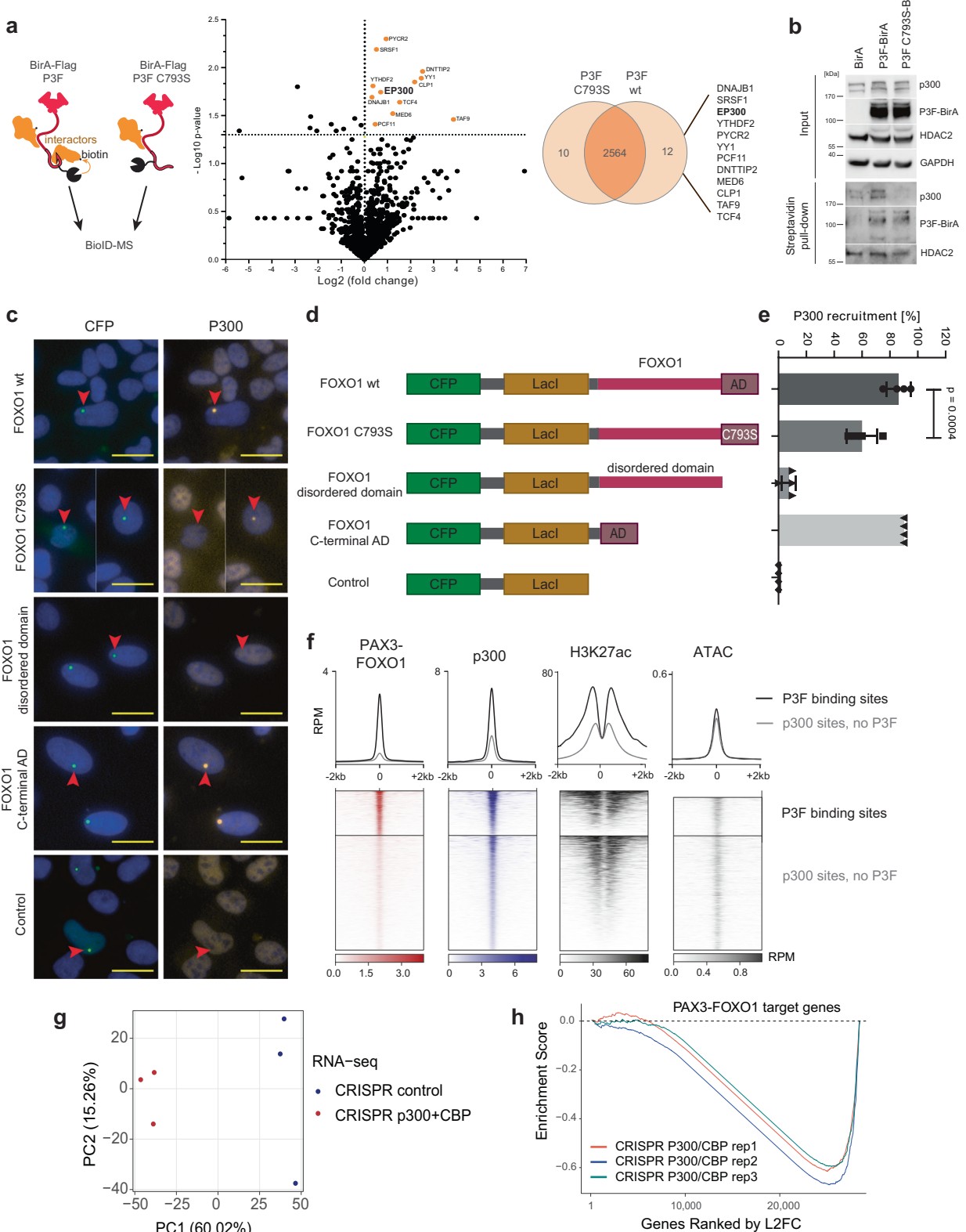

by P3F is the basis for selective effects of the tested inhibitors on expression of P3F target genes.

We selected dCBP1 for further validation with a set of three PDX-derived primary FP-RMS models in comparison to RH4 cells. 24 h of treatment efficiently reduced p300 as well as MYCN levels in all of these models at low nanomolar concentrations (Fig. 5i). While P3F itself is less affected both at the protein and the mRNA levels after

6 hours, mRNA levels of different P3F target genes are strongly downregulated (Supplementary Fig. S7e and Fig. 5j). Treatment of cells for 6 days was found to induce a drastic reduction of cell numbers without induction of cell death (Fig. 5k). Instead, a strong G1 cell cycle arrest was detected after p300 degradation in all models (Fig. 5l). Additionally, H3K27ac ChIP-seq performed in RH4 cells treated with DMSO and dCBP1 (6 h treatment) shows a clear reduction in

**Fig. 4 | Mutation of C793 affects binding of P3F to CBP/p300. a** Left panel, schematic of design for BioID experiment. Center panel, volcano plot depicting results of the BioID experiment using HEK293T cells transfected with P3F-BirA or C793S mutant P3F-BirA for identification of interactors. Log2 fold change of total spectral counts measured by mass spectrometry from the comparison of wildtype and C793S mutant P3F is shown (*n* = 3 independent experiments, unpaired two-tailed T-test). Right panel, Venn diagram depicting differences in the interactome of wildtype P3F versus C793S mutant P3F. The 12 proteins specific for wildtype P3F are indicated on the right. **b** Validation of BioID-MS results by Western Blot. BioID labeling was performed in 293T cells transfected with BirA or wildtype or C793S mutant P3F-BirA fusion protein constructs. Indicated proteins were detected in cell lysates for detection of input levels (upper panels) and in streptavidin pull-downs for evaluation of biotinylation levels (lower panels). One representative experiment from *n* = 2 is shown. **c** Representative pictures from the p300 recruitment assay performed with Lac-U2OS cells transfected with indicated LacI-CFP-FOXO1 fusion constructs. The CFP signal indicates the location of the LacI-CFP fusion; presence of p300 was determined by immunofluorescence staining. Pictures from one out of four independent experiments are shown. Scale bar, 20 μm. **d** Scheme displaying the LacI-CFP-FOXO1 constructs used for the recruitment assay shown in c. **e** Quantification of the p300 recruitment assay. Presence or absence of p300 signal was determined for each CFP dot and counted (*n* = 4 independent experiments, plotted as means ± SD, one-way Anova, Tukey's multiple comparisons test). **f** Heatmap of p300, P3F, H3K27ac ChIP-seq and ATAC-seq signal. **g** RNA-seq analysis of RH4 cells after double knockout of p300 and CBP. Total RNA was isolated from p300/CBP double knockout and empty vector control cells three days after sgRNA transduction (*n* = 3 independent experiments for each knockout). PCA plot performed with normalized and log-transformed count data is shown. **h** Gene set enrichment plot for RNA-seq after knockout of p300 and CBP, resulting in selective downregulation of P3F target genes. Source data are provided as a Source Data file.

acetylation at target genes like PIPOX (Fig. 5m) and genome-wide, particularly at sites with both P3F and p300 binding (Fig. 5n and Supplementary Fig. S7F). Taken together these results show that inhibition of the p300/CBP axis might be an efficient way to interfere with P3F activity in FP-RMS.

## Pol2 clusters at PAX3-FOXO1 target genes require CBP/p300 for proper looping distribution

We next aimed to further pinpoint the mechanism of action by which p300/CBP degradation interferes with P3F-induced transcription. Histone acetylation mediated by HATs including p300 is required for Pol2 to access histone-occupied chromatin[18], and Pol2 forms condensed clusters at super-enhancers[19]. To evaluate whether P3F was shaping clusters of Pol2 that potentially require CBP/p300, we built Pol2 connectivity maps using HiChIP[20]. Using Pol2 contacts we derived ranked Pol2 clusters sorted by their degree of connections and total densities (Fig. 6a). P3F-dependent core regulatory TFs (e.g., *SOX8, MYOD1*) and other targets (e.g., *PIPOX*) were involved in high density cluster regions (Fig. 6a), in which the probability of contacts linearly increased with P3F occupancy (Fig. 6b). 3D connectivity by H3K27ac tracked the topological patterns observed for RNA Pol2, with locus-specific enhancer-promoter interactions within the high-density cluster regions clearly evident (the *MYOD1* locus is shown as a representative example in Fig. 6c). We observed that nearly a third of all these high-density clusters mapped to genomic loci that contained P3F ChIP-seq peaks (Fig. 6d). Indeed, transcriptional sensitivity to dCBP1 was greatest among genes in Pol2 clusters with more P3F binding sites (Fig. 6e).

The hyper-dependence on CBP/p300 seen for genes within P3F-bound Pol2 clusters motivated a direct comparison via AQuA-HiChIP for Pol2 organization in 3D before and after dCBP1 treatment. Pol2 colocalization at P3F and p300 binding sites was decreased, whereas Pol2 at the TSS sites increased following treatment with dCBP1 (Fig. 6f, g). The 374 clusters with P3F had 7.7 Pol2 loops per cluster on average, 1.4x P3F peaks on average, and an average size of 73 kilobases, and Pol2 connected P3F to its target genes as shown with aggregate peak analysis (APA, Fig. 6f). However, changes in acetylation did not always follow changes in Pol2 accumulation. Comparing Pol2 HiChIP to H3K27ac HiChIP, we observed that following dCBP1 treatment H3K27ac contact signal is overwhelmingly lost whereas RNA Pol2 contact signal appears to increase and decrease in near equal measure across sites (Fig. 6h). It was this discrepancy that led us to explore where Pol2 was moving in the genome.

## Degradation of p300/CBP shifts Pol2 activity from CpG-poor to CpG-rich regions in the chromatin

Splitting our sites of Pol2 contact gain and loss by distance from the TSS, we identified a clear gain at regions within 5kB of gene promoters, a non-intuitive finding given the demonstrated downregulation of

genes core to driving FP-RMS (Fig. 7a). Further analysis showed that, while Pol2-loss sites did contain cis-regulatory sequences (P3F motifs, MYOD and MYOG motifs, and SOX8 motifs) (Fig. 7b) for the expected genes, Pol2-gained sites were at NFY motifs and frequently (83%) within CpG islands (Fig. 7c), similar to the finding in cells with C793 mutant P3F (Fig. 3i). While the former is interesting given previous categorization of NFY as a P3F activator[21], the latter is consistent with a model in which Pol2 is strongly dependent upon acetylation for binding to non-CpG dense chromatin. Subcategorized further, those regions which experienced the most loss of contacts among Pol2 clusters were those which contained both p300 and P3F binding at the ends of their loops (Fig. 7d) and among clusters with the most CpG island occupancy, there are few, if any, P3F binding sites (Fig. 7e). Taken together, our data show how CpG islands are capable of accommodating Pol2 in the absence of histone acetyl transferases, a feature conferred by the rigidity of CpG rich sequences that destabilizes histone wrapping and promotes accessibility[20].

## Pol2 cluster collapse at P3F regulated genes

Based on these findings we wondered about the precise point of Pol2's failure to transcribe upon loss of acetylation. We initially hypothesized a defect in pause-release, as has been reported for BRD4 (ref. 22). We therefore examined the ratio of Pol2 binding to the pause site, relative to the body of the gene, across the CR TFs that lost almost all of their transcriptional output. We found that while globally many genes have gained Pol2 at the promoters, the strongly downregulated CR TFs had an inconsistent mix of increased or decreased traveling ratio (the proportion of Pol2 in the pause-site compared to elongating Pol2 in the body of the gene). This mixed effect prevented us from attributing the defect in transcription to a failure to elongate (Supplementary Fig. S8). We then noticed that the gain in the body of the genes and the promoter proximal regions did not extend past the transcriptional end site (TES) for these halted genes. Thus, we designed a new metric to capture this behavior we termed the Pol2 UnLoading Ratio (PULR), the ratio between Pol2 signal anywhere in the gene prior to the TES divided by the Pol2 signal after the TES (Supplementary Fig. S9). Using this metric, we measured an increased Pol2 UnLoading Ratio that was consistent across downregulated CR TF genes, indicating incomplete transcription termination.

We next examined trends in the altered Pol2 looping relationship induced by CBP/p300 removal across downstream target genes of P3F. Pol2, P3F, p300, and H3K27ac pile up at near these genes, resulting in the formation of super enhancers (Supplementary Fig. 10a). From Pol2 AQuA-HiChIP data[23], we found that P3F target genes including *SOX8, MYOD1, PIPOX* and *FGFR4* were experiencing a loss of Pol2 looping to their enhancer elements and a strong, central gain of Pol2 near their TSS after dCBP1 treatment (in general Fig. 8a; in specific Fig. 8b, Supplementary Fig. S10b and S10c). We termed this phenomenon Pol2 "cluster collapse." We further discovered that

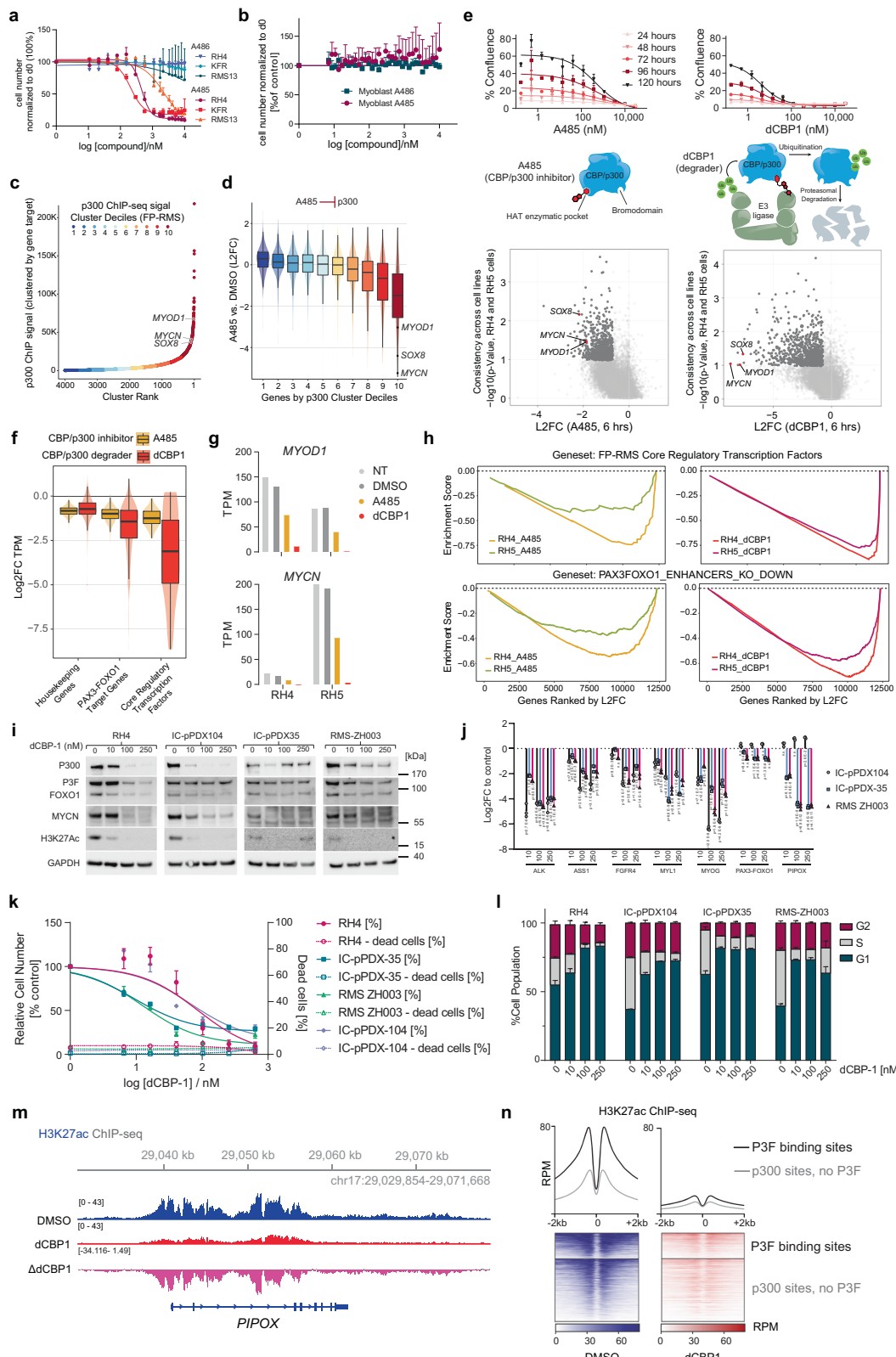

the large concentration of Pol2 was collecting at nearby promoters with CpG islands. At genes lacking CpG islands, such as the P3F target gene *PIPOX*, loss of Pol2 at both enhancers and the promoter was observed, while Pol2 piled up at a nearby CpG-promoter of another gene. Genome-wide, the loops most gained were between CpG-promoters. By contrast, the loops most eroded involved enhancers without CpG islands (Fig. 8c). Additionally, we performed Pol2

HiChIP on an RH4 cell line where P3F was engineered[24] to encode a protein tag FKBP12(F36V) after treating it with the degrader dTAG-47 (Supplementary Fig. 10d) and saw a similar collapse of Pol2 signal at long-range contacts (25 Kb to 3 Mb in distance, Fig. 8d). Taken together, our data suggests that protein-protein interactions of P3F with CBP/p300 lead to acetylation of chromatin that facilitates Pol2 recruitment, and that inhibition of the CBP/p300-P3F interaction

**Fig. 5 | CBP/p300 are required for RMS growth and the activation of P3F target genes. a, b** Relative cell number after treatment of cells with A485 and A486 for 6 days, as determined by high content microscopy (*n* = 3 independent experiments, means ± SD). **c** ChIP-seq clusters of p300 in enhancers in RH4 cells. **d** L2FC of mRNA in RH4 cells treated with 10 μM A485 for 6 h. Genes are ranked according to p300 cluster decile (*n* = 1 independent experiment, box plots of median and quartiles, whiskers showing 1.5 × inter-quartile ranges). **e** Upper panels, dose-response curves of RH5 cells treated with A485 and dCBP1 over 5 days as determined by Incucyte microscopy (n = 2 independent experiments, means ± SD). Middle panels, schemes depicting the mechanism of action for both drugs. Lower panels, plots comparing RNA-seq data from cells treated with A485 and dCBP1 for 6 h (*n* = 1 independent experiment, Welch's *t*-test). **f** Degree of downregulation of indicated gene classes after CBP/p300 inhibition (*n* = 1 independent experiment; box plots of median and quartiles, whiskers showing 1.5 × inter-quartile ranges).

**g** TPM values of *MYOD1* and *MYCN* after drug treatment in RH4 and RH5 (*MYCN*-amplified) cells. **h** GSEA comparing the degree of downregulation of core regulatory transcription factor genes and genes within P3F enhancer TADs upon P3F silencing via shRNA. **i** Western blot detection of indicated proteins from indicated cells after dCBP1 treatment for 24 h. One representative blot from n = 3 is shown. **j** mRNA levels in indicated cells after 24 h of dCBP1 treatment. L2FC to DMSO is shown (*n* = 3 independent experiments, means ± SD, Tukey's multiple comparisons test). **k** Relative cell number and percentage of dead cells in indicated cells after dCBP1 treatment for 6 days (*n* = 3 independent experiments, means ± SD). **l** Cell cycle analysis of indicated cells after dCBP1 treatment for 24 h (*n* = 3 independent experiments, means ± SD). **m** H3K27ac ChIP-seq data for the *PIPOX* locus in RH4 cells treated with DMSO or dCBP1. Delta track shows control subtracted from dCBP1. **n** H3K27ac ChIP-seq signal in RH4 cells before and after treatment with dCBP1 for 6 h. Source data are provided as a Source Data file.

collapses these Pol2 clusters, halting P3F target gene expression (Fig. 8e).

## Discussion

Most FP-RMS cells rely on expression of the oncogenic fusion transcription factor P3F. The identification of functional domains in oncogenic TFs like P3F represents a high-priority pool of molecular targets for therapeutic intervention. Here, using a combination of CRISPR domain screening, amino-acid substitution functional studies and dCas9-based base editing, we pinpointed C793 within the alpha-helical coil region of the P3F AD as a new critical residue inside a newly appreciated domain for the function of this oncoprotein. While the C-terminal domain previously has been shown to be relevant for P3F activity in heterologous cell lines[25,26], we demonstrate here that mutation of C793 into a serine residue results in loss of expression of P3F target genes, such as *PIPOX*, *FGF8* and *ASS1*, in its endogenous context. Similar to genetically induced loss of P3F expression, we observed strong induction of differentiation as monitored by induction of structural myogenic genes. The contribution of the large disordered part of FOXO1, which may be involved in a condensation process to form phase separated droplets[27], to the activity of P3F, remains unknown. However, small amino acid changes introduced in our CRISPR domain screen might not be sufficient to reveal functional aspects of this disordered domain.

We were able to attribute P3F's function in part to its ability to recruit p300/CBP via C793. Importantly, this cysteine is conserved among all members of the FOXO family and has been shown to be involved in the functionality of wild-type FOXO proteins at multiple levels[10,28,29]. Under certain redox conditions, the corresponding cysteine (C477) in the AD of FOXO4 was found to form disulfide bridges with cysteines of several interacting proteins including p300 to generate a stable, covalent interaction[28,29]. For P3F, we could show that interaction with individual p300 domains such as the KIX domain is rather weak, while combinations of these domains including the TAZ1, KIX, ZZ-TAZ2 and IBID domains or full length p300 showed a stronger recruitment. Such multiple binding has already been described in case of the p53-p300 interaction[16]. Altogether, these data suggest that the mode of interaction between the AD of P3F and p300 could involve both stable and multivalent weak interactions.

Apart from p300, our BioID experiment identified other potential interactors of P3F that depend on C793, including MED6, a component of the mediator complex and TAF9, a component of TFIID, that supports the assembly of the pre-initiation complex (PIC). Recent studies demonstrated that p300/CBP, besides acetylation-dependent BRD4 recruitment, also promotes assembly of the PIC by acetylation of sites subsequently bound by the bromodomain protein TAF1, another component of TFIID (ref. 30). Accordingly, acute transcriptional consequences of decreased histone acetylation levels at promoter regions after p300 inhibition are due to both loss of Pol2 binding and reduction of pause release mediated by BRD4 (refs. 30,31). This suggests that

MED6 and TAF9 follow p300 recruitment to P3F sites, reflecting the ongoing establishment of the transcriptional machinery. Along these lines, our data also revealed a dramatic effect of the P3F-p300/CBP axis on Pol2 hubs. Pol2 clustering is an important determinant for the transcriptional output of genes. More than 80 molecules of Pol2 are estimated to condense in 300–400 nM droplets[32], forming liquid-like condensates at super-enhancers[19], and it has been established that the AD of TFs drives this[33]. Our data showing the consequences of near complete degradation of p300 by the recently advanced CBP/p300 degrader dCBP1 (ref. 17) confirms this mechanism and reveals a reduced recruitment of Pol2 to P3F binding sites. The more P3F sites overlapped Pol2 clusters, the more downregulated these clusters were after a 6-hour treatment with dCBP1. In contrast, the effects upon C793 mutation are more subtle. The mutation does not affect recruitment of Pol2 to P3F sites, but rather leads to its trapping at these sites, suggesting that under conditions of reduced but not completely absent p300, Pol2 pause release is more affected than recruitment. This data highlights the importance of CBP/p300 for Pol2 clustering at P3F target genes in FP-RMS and also confirms the previous finding that the ability of enhancers to activate genes-at-a-distance depends on CBP/p300 (ref. 34).

After both p300 degradation and C793 mutation we also detected a striking accumulation of Pol2 at CpG islands of CpG island-containing promoters of genes within Pol2 transcriptional clusters, which is accompanied with a collapse of long-range Pol2 contacts. These genes, despite abundant Pol2 in the promoter and gene body, lack binding of Pol2 past the transcriptional end site, and potentially fail to produce mature RNA transcripts. RNA Pol2's retreat to CpG islands upon CBP/p300 degradation is likely a result of these islands remaining accessible despite a lack of acetylation on nearby histones. The loss of acetylation collapsing Pol2 into CpG islands on chromatin is in stark contrast to the effect of hyper-acetylation on Pol2 that removes Pol2 from both enhancers and promoters of these same genes[13], indicating that the dynamic opposition of HATs and HDACs at Pol2 clusters is striking a critical balance of acetylation[35]. These data paint a mechanistic picture of a fusion-oncogene as an enabler of 3D Pol2 hubs via long-range *cis*-recruitment of histone acetylation enzymes. This highlights the fusion oncoprotein P3F as a prominent guiding force for the strength of Pol2 clustering, and for the interaction frequency between enhancer-bound Pol2 and promoter-bound Pol2 in FP-RMS cells.

Interestingly, we did not detect an influence of p300/CBP on DNA accessibility, which agrees with previous studies showing that loss of chromatin accessibility after p300/CBP inhibition is small[30,36]. Furthermore, 3D chromatin organization as measured by Hi-C has been shown to be largely independent of p300 activity[37]. Taken together this suggests that chromatin opening is regulated at a more upstream level, before the recruitment of p300/CBP to regulatory sites in the genome.

Overall, our data characterizes CBP/p300 as an important amplifier of P3F-mediated gene expression and demonstrates that

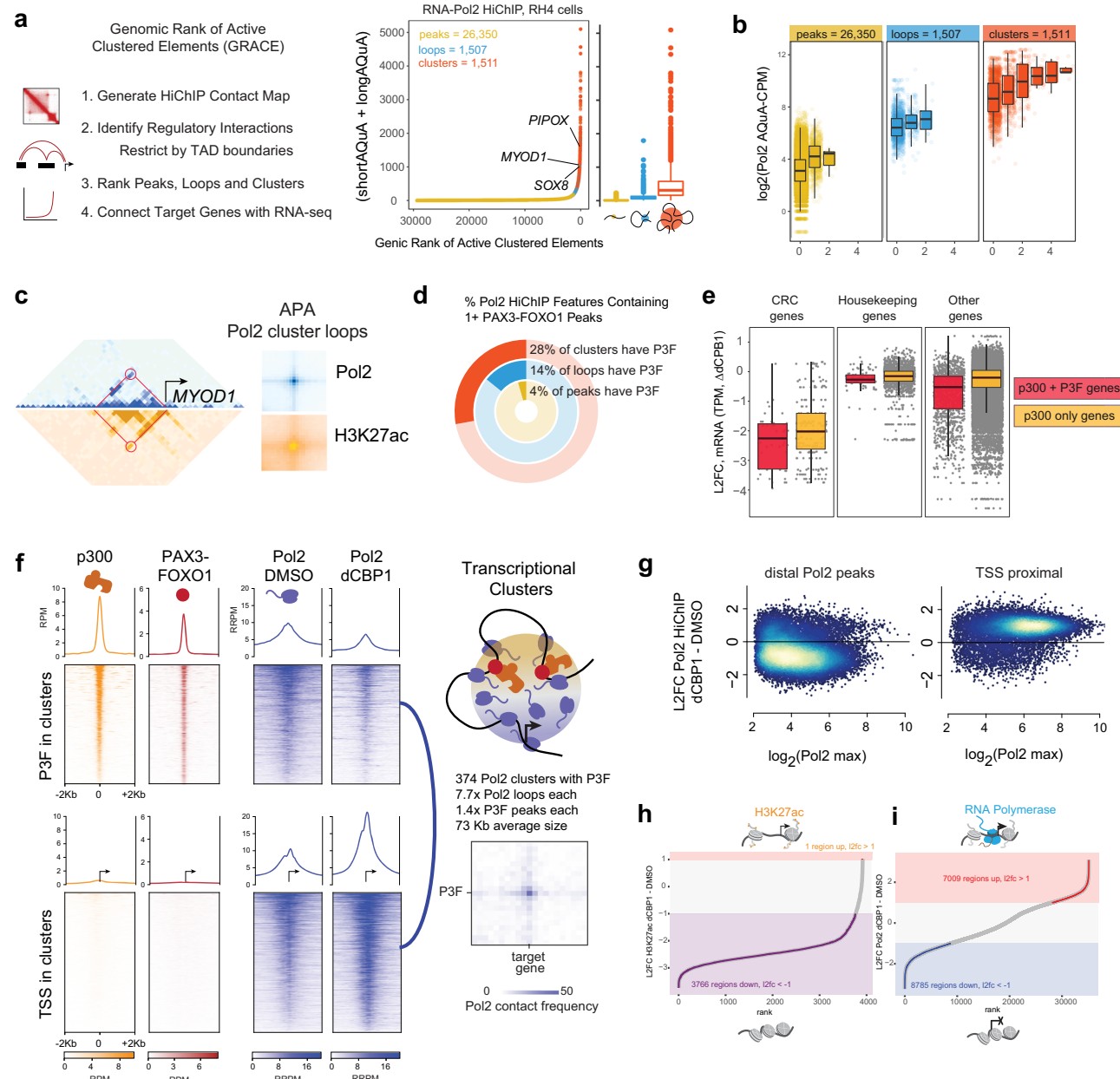

**Fig. 6 | p300 is required for Pol2 clusters connecting P3F to its targets.**
**a** Number of Pol2 HiChIP contacts (short and long AQuA-normalized contacts, counts-per-million (CPM)) in RH4 cells plotted for all HiChIP features ranked along the X-axis by increasing HiChIP signal using the GRACE algorithm. HiChIP features are categorized by number of HiChIP contacts, with peaks, loops, and clusters having 0, 1, and 1+ loops, respectively. Box plots of median and quartiles, whiskers showing 1.5 × inter-quartile ranges. **b** 3D connectivity as function of overlapping P3F peaks for Pol2 peaks, loops, and clusters. **c** Left, HiChIP contact maps (5 kb bins) for Pol2 (blue) and H3K27ac (gold) in RH4 cells at the *MYOD1* locus (red circle, example cluster loop). Right, Aggregate Peak Analysis (APA) plots of all Pol2 loops from clusters. Box plots of median and quartiles, whiskers showing 1.5 × inter-quartile ranges. **d** Percentages of Pol2 HiChIP features containing 1 or more P3F peaks (from independent ChIP-seq experiment). **e** Pol2 HiChIP features plotted by the L2FC in expression of indicated gene classes associated with the contacts in

RH4 cells following 6 h dCBP1 treatment. Each group is split between features which associate TSS to p300 sites with or without P3F. (*n* = 1 independent experiment; box plots of median and quartiles, whiskers showing 1.5 × inter-quartile ranges. **f** ChIP-seq signal in RH4 cells for p300 and P3F, and HiChIP for Pol2 before and after 6 h dCPB1 treatment for all Pol2 cluster anchor constituents that contain P3F ChIP-seq peaks (top panels), and at the TSS for genes contained in Pol2 HiChIP clusters (bottom panels); scheme of Pol2 HiChIP cluster containing p300, P3F, and Pol2 (top right). Pol2 HiChIP APA plot showing the 3D connectivity of one distal P3F-bound enhancer and its putative target TSS (bottom right). Blue line, positions in the left-adjacent 2D heatmaps. **g** M-A scatter plots for L2FC of Pol2 HiChIP density at Pol2 peaks distal and proximal to TSS after 6 h dCBP1 treatment of RH4 cells. **h,i** Rank plots of RH5 H3K27ac HiChIP (**h**) and RH4 Pol2 HiChIP (*i*) regions based on L2FC of AQuA RRPM between DMSO and dCBP1 treatments. Schematics show active (top) and inactive (bottom) chromatin.

recruitment of p300/CBP to its target genes is a very important function of P3F. This complements existing data of the mode of action of P3F in RMS cells. Chromatin binding of P3F has been shown to be affected by the chromatin remodeler CHD4 (ref. 6). The following recruitment of p300/CBP to these sites then promotes PIC assembly, leading to Pol2 super clustering as seen in this paper. BRD4

activity at these sites is also required, but for RNA Pol2 cluster flexibility that enables transcription of CR TFs in RMS, and BRD4 degradation results in a failure of Pol2 to unload past the TESR of CR TFs[38]. Interference with any of these steps affects P3F-driven gene expression and therefore might have therapeutic promise for patients with FP-RMS.

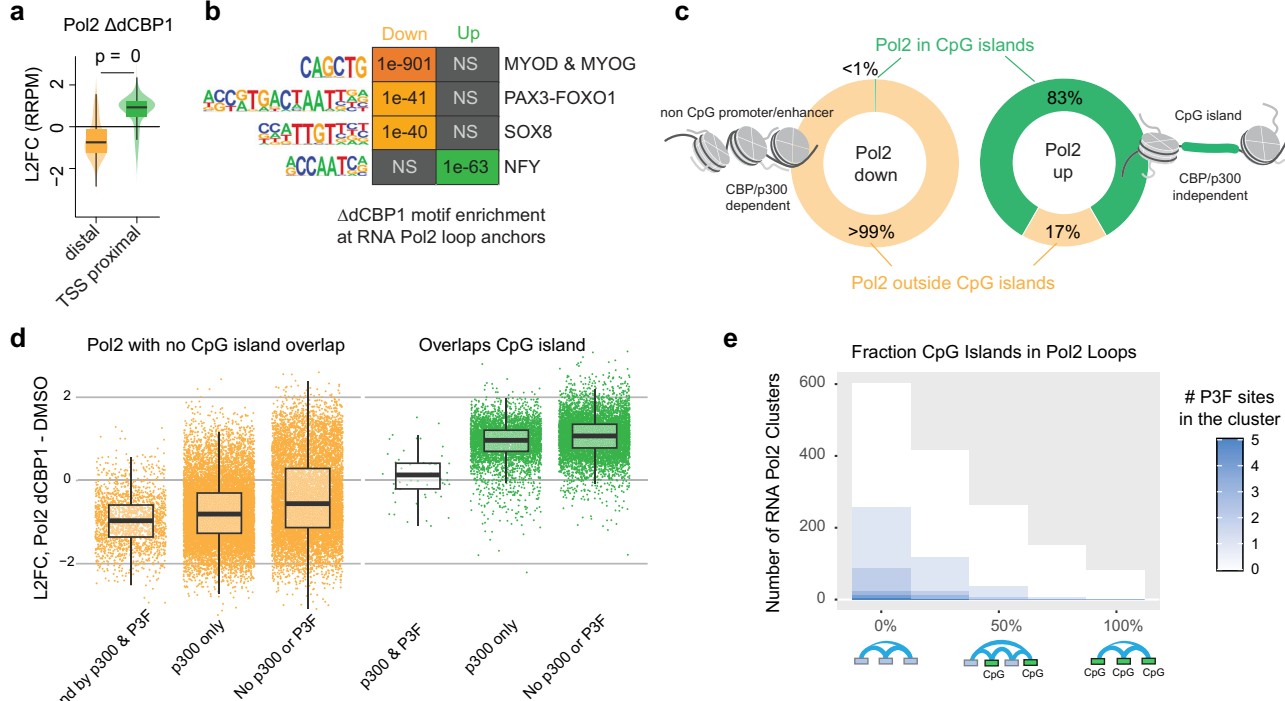

**Fig. 7 | CpG islands determine response to CBP/p300 degradation. a** L2FC of RNA Pol2 binding (RRPM) proximal (<=5kB) or distal from the gene TSS at 6 h after dCBP1 treatment (n = 1 independent experiments; Welch's *t*-test (p = 0.0) was performed between samples; Box plots of median and quartiles, with whiskers showing 1.5 × inter-quartile ranges). **b** After 6 h dCBP1 treatment, the motifs of the down-regulated peaks indicate their positioning at key transcription factors (TFs) of FP-RMS, *MYOD*, *MYOG*, *P3F*, and *SOX8*. Additionally, an increased motif enrichment of RNA Pol2 is observed at *NFY*, a promoter-binding TF that contributes a phenotype similar to that of P3F. NS, not statistically significant. **c** Percentage of up- and downregulated RNA Pol2 peaks at CpG islands of CBP/p300 independent and

dependent genes. <1% of all downregulated RNA Pol2 peaks are located at CpG islands of CBP/p300 dependent gene promoters. **d** Boxplots showing the Log2(-Fold Change) in the Contact Signal (RRPM) of RNA Pol2 loops between DMSO and dCBP1 treated RH4 cells, split by p300 and P3F occupancy and overlap with CpG islands (n = 1 independent experiment; Box plots of median and quartiles, with whiskers showing 1.5 × inter-quartile ranges). **e** A histogram showing the distribution of RNA Pol2 clusters from RH4 HiChIP based on the overlap between a cluster's constituent loops and CpG islands. The number of P3F binding sites overlapping the ends of the loops in the cluster is shown in blue.

The broad functionality of p300/CBP might speak against these proteins as direct targets for such a therapy. Importantly however, two p300/CBP bromodomain inhibitors CCS1477 and FT-7051 are currently in clinical trials for cancer patients, and preliminary data showed good tolerance[39]. Furthermore, we could show that CBP/p300 is bound to an even greater degree at P3F sites compared to non-P3F bound enhancers. Accordingly, CBP/p300 degradation caused efficient halting of P3F target genes and the FP-RMS core regulatory circuitry, while in comparison downregulation of housekeeping genes was only limited. This is in line with recent data showing that p300/CBP inhibition selectively downregulates cell type-specific genes[30] and suggests that there might be a therapeutic window available for p300/CBP inhibitors. Nevertheless, since we detected only cytostatic effects in FP-RMS cells, p300/CBP-inhibitors might not represent the first choice. Instead, compounds that interfere with the recruitment of p300 by the FOXO1 part of the fusion and mimicking genetic inactivation of the AD of P3F, might be more promising. Interestingly, a compound with such a mechanism of action has recently been described for wildtype FOXO1 (ref. 40). Potentially, C793 and its surrounding domain could also be a target for an inhibitory compound. Whether C793 is reactive enough to serve as target for a covalent inhibitor similar to inhibitors developed for other oncogenes[41] represents an interesting question for future studies.

Additionally, while prior work identified that loss of FOXF1 (ref. 42) and NFY (ref. 21) can induce myogenic differentiation specifically in FP-RMS, our evaluation of the downstream effects of P3F mutation and degradation show reduced expression of both genes

and, in the case of NFY (Fig. 7b), imply changes in Pol2 UnLoading Ratio (PULR, Supp. Fig. 9). While FOXF1 DNA-binding was associated with p300 activity, FOXF1 binding near P3F sites in active enhancers may be correlative and the absence of specific evidence that FOXF1 recruits p300 suggests that P3F may still be the core driver of oncotranscription. P3F activity and therefore P3F AD function could be the most potent and specific target for reducing FP-RMS proliferation.

## Methods
### Cell lines
FP-RMS cell lines RH4, RH5 (both provided by Peter Houghton, Greehey Children's Cancer Research Institute, San Antonio, Texas, USA), RMS (Janet Shipley, Sarcoma Molecular Pathology, The Institute of Cancer Research, London, UK), KFR (Jindrich Cinatl, Frankfurter Stiftung für krebskranke Kinder, Frankfurt, Germany), myoblast cell line KM155C25Dist (provided by Vincent Mouly, Institut de Myologie, Paris, France), RH4-PAX3-FOXO1-FKBP12^F36V (provided by Dr. Kristy R. Stengel, Albert Einstein College of Medicine, New York, NY, USA) as well as HEK293T cells and C2C12 cells (both purchased from ATCC), were maintained in DMEM (Sigma-Aldrich) supplemented with 10% FBS (Thermo Fisher Scientific, LuBioScience), 2 mM L-glutamine, and 100 U/ml penicillin/streptomycin. PDX-derived cell models IC-pPDX-104, IC-pPDX-35 (both established from PDX tumors provided by Didier Delattre, Institute Curie, Paris, France) and RMS-ZH003 (established from a PDX tumor generated in Zurich) were cultured on a thin Matrigel coating and maintained in advanced DMEM/F12 medium

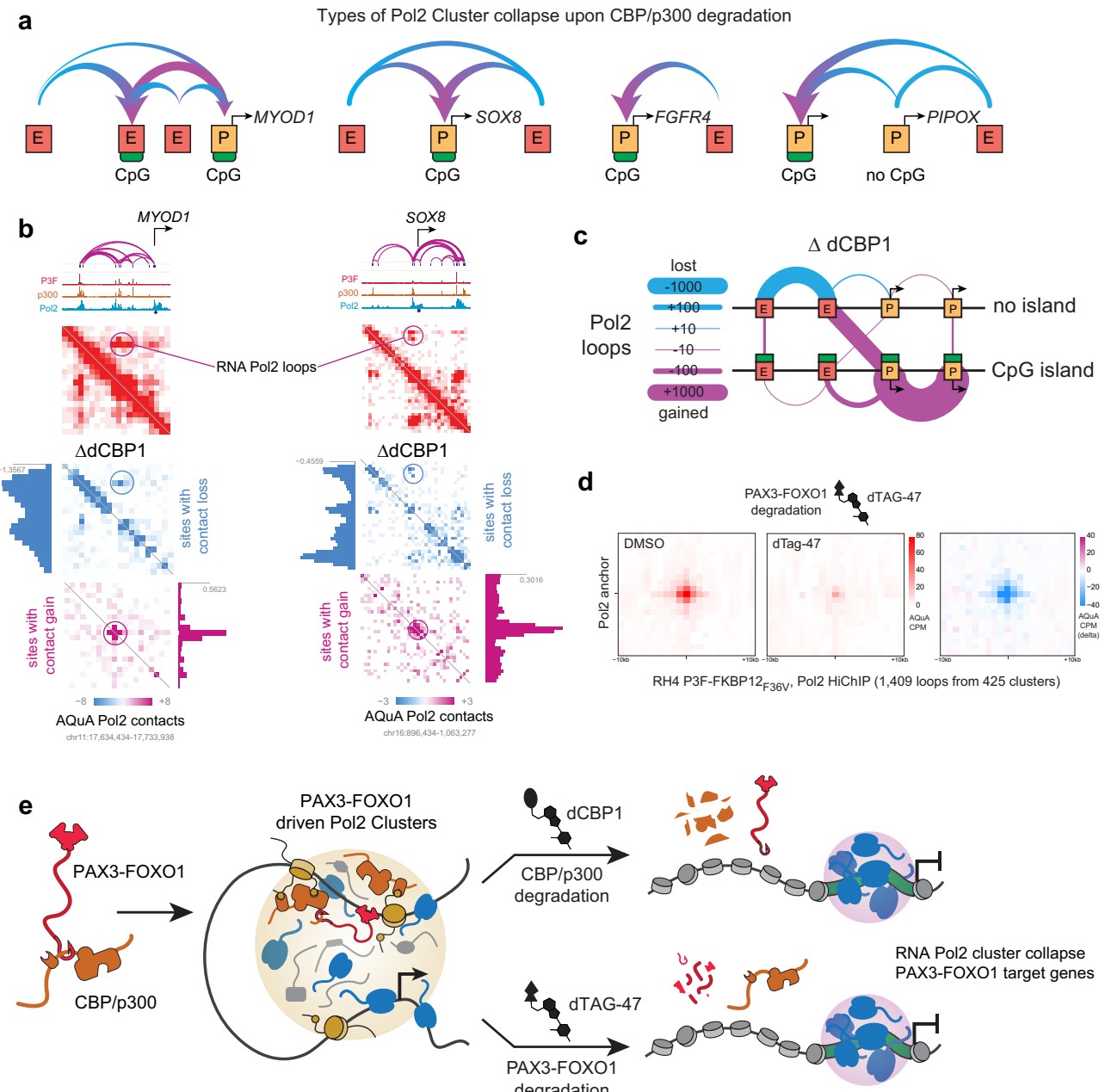

**Fig. 8 | RNA Pol2 clusters collapse, migrating to promoters with CpG islands.**
**a** Schematic patterns describing the various types of RNA Pol2 cluster collapse. In these instances, RNA Pol2 migrates from enhancers and promoters without CpG islands and collects at promoters with CpG islands across *MYOD1*, *SOX8*, *FGFR4* and *PIPOX*. **b** 3D contact maps of *SOX8* and *MYOD1* showing AQuA-CPM RNA Pol2 contacts. After 6 h dCBP1 treatment, there is an observed loss of long-range contacts with a gain of short-range contacts at a central location. **c** Schematic map of RNA Pol2 migration between promoters and enhancers with and without CpG island. The RNA Pol2 contacts between enhancers lacking CpG islands were mainly

lost, whereas the contacts between the promoters having CpG islands were highly gained. The enhancers lacking CpG islands gained contacts to CpG containing promoters. **d** Aggregate Peak Analysis (APA) plots for RNA Pol2 HiChIP in RH4 cells edited to express PAX3-FOXO1-FKBP12(F36V) treated with DMSO (left), dTAG-47 (center), and the delta of dTAG-47 compared to DMSO (right). AQuA CPM is represented as bins of +/− 1 kilobases at Pol2 anchors, **e** Depiction of the super-cluster disruption model. CBP/p300 entangled with P3F acetylates enhancers to seed RNA Pol2 clustering and dynamic interaction with P3F target genes, which is disrupted when the CBP/p300-P3F driven RNA Pol2 super-clustering is lost.

(Thermo Fisher Scientific, LuBioScience) supplemented with 1xB27 (Thermo Fisher Scientific, LuBioScience), 1.25 mM N-acetyl-L-cysteine (Sigma), 5 μM A83-01 (Tocris Bioscience), 10 μM Y-27632 (Abmole Bioscience), 10 ng/ml bFGF and EGF (both from Peprotech), 2 mM L-glutamine, and 100 U/ml penicillin/streptomycin, as we described previously[43]. All cells were cultured in 5% $CO_2$ at 37 °C. All RMS cell lines were authenticated upon receipt by short tandem repeat (STR) profiling and used for experimentation from frozen stocks. As the KM155C25Dist cell line has not yet been characterized by STR profiling,

negative matching with all available cell lines in the database was used for verification. All cells were regularly tested for mycoplasma contamination by a PCR-based assay. Derivative cell lines RH4 shP3F, KFR shP3F as well as RMS13 shP3F were generated previously[5].

**Small molecule compounds**
All small molecules were dissolved in DMSO. dCBP1 was provided by Christopher Ott (Massachusetts General Hospital Cancer Center). A485 was provided by Adam Durbin (St. Jude Children's Research

Hospital) or purchased (Selleckchem), while the control compound A486 was provided by the Structural Genomics Consortium. dTAG-47 was purchased (Tocris Bioscience #7530).

### sgRNA design and cloning

All sgRNAs targeting *P3F* as well as the control guides targeting the first exon of *FOXO1* or exon 8 of *PAX3* were manually designed. Guides targeting *p300* or *CBP* were designed with the GPP sgRNA designer webtool (https://portals.broadinstitute.org/gpp/public/analysis-tools/sgrna-design). Each guide was cloned individually by annealing two DNA oligos and ligating the double stranded product into the sgRNA scaffold of the Esp3I digested sg_shuttle destination vector. The control guide targeting the AAVS1 locus was cloned into the sg_shuttle plasmid containing BFP as selection marker (sg_shuttle_BFP, derived from sg_shuttle_RFP657), while all other sgRNAs were cloned into a similar plasmid containing RFP657 as selection marker (sg_shuttle_RFP657¸ Addgene #134968). For improvement of transcription efficiency, a G nucleotide was added at the 5′-end of all sgRNAs that did not start with a G. All sgRNA sequences used in this study are shown in Supplementary Data S1.

### Production of lentiviral particles and transduction of cells

Lentiviral particles were produced in HEK293T cells by calcium phosphate-based co-transfection of the transfer plasmids together with lentiviral packaging plasmids pMD2.G (Addgene #12259) and psPAX2 (Addgene #12260). 24 h after transfection, medium was replaced and virus was harvested 48 h later. Viral supernatants were concentrated using Amicon® Ultra 15 ml columns with a cut-off of 100 kDa (Millipore). 80,000 cells grown in 24-wells were transduced with 2 µl viral concentrates diluted in 0.5 ml medium supplemented with 8 µg/ml polybrene (Sigma Aldrich #107689).

### Domain screen

RH4 cells stably and uniformly expressing Cas9 were obtained by transducing RH4 cells with the pL40C_PGKintron_Cas9_Green expression vector coding for both Cas9 and mNeonGreen (Addgene #134966), followed by fluorescence activated cell sorting for a high mNeonGreen positive population. Two days after transduction of RH4-Cas9 with sgRNA plasmids, RFP and BFP populations were mixed in a 1:1 ratio. Relative proliferation of RFP and BFP populations was determined by flow cytometry at days 2 and 12. Effects of the guides was determined by calculation of depletion of RFP+ cells at day 12 vs day 2.

### Flow cytometry

After detachment from dishes, cells were washed once with PBS and fixed with 0.5% paraformaldehyde in PBS (Roth, P087.3) for 5 min at room temperature. Cells were then transferred through a cell strainer cap into round-bottom polystyrene test tubes and analyzed on a BD LSRFortessa instrument equipped with the BD FACSDiva 8.0.1 software. Analysis of data was then performed with the FlowJo V10.8.1. software.

### Luciferase reporter assay

A pGL4.19 (Promega) based reporter construct in which firefly luciferase is under control of a P3F-responsive element from the *ASS1* gene (NCBI36/hg18, chr9: 132344452-132344864) was generated to measure P3F transactivation activity. A plasmid with a mutant P3F-binding site was used as control. For the assay, 10,000 HEK293T cells were seeded per 96-well in 50 µl medium. 24 h later cells were transfected with wildtype or mutant reporter plasmids in combination with a plasmid driving expression of the Renilla luciferase for normalization of transfection efficiency as well as different amounts of P3F expression constructs (1–100 ng). 24 h later, medium was changed, followed by measurement of Renilla and Firefly luciferase activity 48 h after

transfection using the Dual-Glo® luciferase assay system (Promega E2920) according to the protocol of the manufacturer.

### Site-directed mutagenesis

Site-directed mutagenesis was performed using the GeneArt Site-Directed Mutagenesis System (ThermoFisher A13282) according to the protocol of the manufacturer. Mutagenesis primer pairs used are listed in Supplementary Data S4.

### *PAX3-FOXO1* gene editing

For site-directed mutagenesis of C793 in endogenous *PAX3-FOXO1*, an adenine base editor approach was used. The coding sequence of the adenine base editor ABE8e from plasmid NG-ABE8e (Addgene #138491) was cloned into the lentiviral plasmid pL40C_pGKintron_Cas9_Green (Addgene #134966), in frame with P2A and mNeon-Green and replacing Cas9. RH4 cells were then stably transduced with this plasmid and a uniform population with robust mNeonGreen expression was selected by FACS (RH4-ABE8e cells). An sgRNA (5′-GTCACAGTCTAAGCGCTCAA) recruiting the base editor to the *PAX3-FOXO1* locus and placing C793 into the base editor window was designed using the webtool "be-designer" (http://www.rgenome.net/be-designer/). As control either an sgRNA located 33 nt further upstream (5′-AATGGGCCTTCTCCACCAGG) or one directed against the AAVS1 locus (5′-GGGGCCACTAGGGACAGGAT) was used. RH4-ABE8e cells were then transduced with this sgRNA cloned into the sg_shuttle_RFP657 (Addgene #134968) or sg_shuttle_BFP plasmid. Five days after transduction, both protein and RNA was isolated from the cells and cDNA was synthesized. The cDNA was then used for amplification of the region around C793 in *PAX3-FOXO1* by PCR using primers For (5′-GAAGACACCTGTACAAGTGC) and Rev (5′-GACACCCAGCTATGTGTCG). The PCR product was gel purified and sequenced by Sanger sequencing using a primer with the sequence 5′-CAATGGCTATGGCAGAATGG. Effects on expression of *ASS1*, *MYH3* and *TNNC2* were measured by qRT-PCR using commercially available assays ("assays-on-demand", ThermoFisher). In parallel, PAX3-FOXO1 protein levels were detected by Western blot.

### Cell proliferation assays

Changes in cell numbers over time were either determined by manual counting or by image based analysis using either an Operetta high content analyzer or an Incucyte ZOOM instrument for automatic measurement of confluence. For manual counting assays, 50,000 cells were seeded per 6 well and cell numbers determined during the following days. For assays with the Operetta high content analyzer, total and dead cells were stained for 15 min with 1 µM Hoechst 33342 (ThermoFisher, #62249) and 1 µM propidium iodide (PI, Calbiochem/Sigma #537060), respectively. After imaging of individual wells with the Operetta instrument equipped with the Harmony 4.9 software, numbers of Hoechst- and PI-positive cells was determined using a script in Harmony 4.9. For monitoring cell confluence by Incucyte ZOOM, cells were plated in 384 well plate format to achieve 3% confluence at the time of the experimental start. Confluence of control (DMSO) wells was monitored until reaching ~40% confluence. Phase-contrast images were taken every three hours; from this, percent phase object confluence was quantified via Incucyte ZOOM software.

### Cell cycle analysis

Cells for cell cycle analysis were grown in 6-well dishes. Floating and adherent cells were combined and resuspended in 50 µl of PBS. 500 µl precooled 70% EtOH (−20 °C) was then dropwise added to the cells under vortexing. After incubation of the cells at −20 °C for at least 2 h, cells were pelleted by centrifugation with 1000 g for 5 min and washed once with PBS. The cell pellet was then resuspended in 500 µl of a solution containing 20 mg/ml propidium iodide (Calbiochem/Sigma #537060), 200 µg/ml RNAse A (Qiagen #1007885) and 0.1% Triton

X-100 and incubated for 30 min in the dark. Cells were then washed once with PBS and analyzed by flow cytometry as described above.

## Quantitative real time PCR

Total RNA was extracted using the RNeasy mini kit (Qiagen, #74106). cDNA synthesis was carried out using the High-Capacity Reverse Transcription kit (Applied Biosystems, #4368814). Quantitative PCR was performed using TaqMan gene expression master mix and Taq-Man gene expression assays (Applied Biosystems) on an ABI 7900UT fast real-time PCR system with the SDS 2.4 software. Each experiment was performed with three technical replicates. Outliers found among technical replicates (SD > 0.5) were removed from the analysis. Cycle threshold (CT) values were then determined with the RQ manager 1.2.1 software (Applied Biosystems). Relative expression levels were calculated using the ΔΔCT method using *GAPDH* expression as reference. Statistical analysis was performed with ΔΔCT values from at least three biological replicates. The following gene expression assays (all from ThermoFisher) were used: ASS1 (Hs01597981), ALK (Hs00608284), GAPDH (Hs02575899), FGFR4 (Hs01106910), CNR1 (Hs01038522), GABRQ (Hs00610921), SEZ6 (Hs01121350), MYH3 (Hs01074230), TNNC2 (Hs00268519(, MYL1 (Hs00984899), PAX3-FOXO1 (Hs03024825), PIPOX (Hs04188864), and GLI2 (Hs01119974).

## Protein isolation and Western blot

Total protein extracts were obtained from cells lysed with RIPA buffer containing 50 mM Tris-Cl (pH 7.5), 150 mM NaCl, 1% NP-40, 0.5% Na-deoxycholate, 1 mM EGTA, 0.1% SDS, 50 mM NaF, 10 mM sodium β-glycerolphosphate, 5 mM sodium pyrophosphate, and 1 mM sodium orthovanadate and supplemented with Complete Mini protease inhibitor cocktail (Roche Diagnostics #11836153001). Protein concentration was measured with the Pierce BCA protein assay kit (Thermo Fisher Scientific, #23225). For Western blots, proteins were first separated using 4%–12% Bis-Tris SDS-PAGE gels (Thermo Fisher Scientific, NP0323) and then transferred to nitrocellulose membranes (Protran, Schleicher & Schuell). After blocking with 5% milk in TBS/0.1% Tween, membranes were incubated with primary antibodies overnight at 4 °C. After washing in TBS/0.1% Tween, membranes were incubated with HRP-linked secondary antibodies. After an additional three was steps in TBS/0.1% Tween and once in TBS, proteins were detected by che-miluminescence using ECL detection reagent or SuperSignal West Femto maximum sensitivity substrate (both Thermo Fisher Scientific, #32106 and #34096, respectively) in a BioRad ChemiDoc Imaging System with the BioRad Image Lab Touch Software 3.0.1.14. The following primary antibodies were used: Anti-p300 (1:1000, CellSignaling #54062), anti-CBP (1:1000; CellSignaling #7425), anti-GAPDH (1:2000; CellSignaling #2118), anti-FOXO1 (1:1000, Santa Cruz; sc-11350), anti-H3K27Ac (1:1000; Active Motif #39685), anti-MYCN (1:1000; Abcam ab16898), anti-HDAC2 (1:1000; CellSignaling #5113) and anti-Flag M2 (1:1000; Sigma F9291). The following secondary antibodies were used: anti-rabbit IgG (1:2000; CellSignaling #7074) and anti-mouse IgG (1:2000; CellSignaling #7076). For processing of the Western blots pictures the Image Lab 6.1.0 software was used.

## BioID experiments

Plasmids for expression of N- and C-terminal BirA-Flag fusion constructs[34] were a kind gift of Philip Knobel (Laboratory for Applied Radiobiology, University Zurich). BirA-Flag/PAX3-FOXO1 fusion constructs were generated by amplification of a revalidated PAX3-FOXO1 cDNA, using primers including restriction sites for AscI (forward) and NotI (reverse), and cloned into N- or C-terminal Bira-Flag backbone vectors. Transient transfection of BirA-Flag/PAX3-FOXO1 fusion constructs or BirA-Flag alone into HEK293T cells was conducted using PEI reagent. Expression as well as subcellular localization of proteins were confirmed by western blot or Immunofluorescence respectively. For Streptavidin immunoprecipitations, 7.5 Mio. HEK293T cells were

plated in a 15 cm plate. The next day, cells were transfected with 12.6 μg of plasmid DNA in presence or absence of 50 μM Biotin. Biotin stock solution (20 mM) was obtained by dissolving 100 mg of powder (IBA, 2-1016-002) in 2.04 ml of NH4OH 28-30% (Sigma Aldrich, ref# 221228). 18 ml of 1 M HCl was added to neutralize the solution (pH-7.5) and stored at 4 °C. 24 h after transfection, cells were harvested by scraping in 1xPBS. After washing once with 1xPBS, cell pellets were resuspended in 1.5 ml Lysis buffer (50 mM TrisHCl pH7.5, 150 mM NaCL, 1 mM EGTA, 1%Triton-X, 0.1% SDS) supplemented with 250U of Benzonase (Novagen, #70664). Lysates were incubated for 1 h at 4 °C under rotation. After brief sonication to disrupt visible aggregates, centrifugation was performed at 16000 g for 30 min at 4 °C. Cleared input samples were incubated together with 75 μl Dynabeads MyOne Streptavidin T1 (Thermo Fisher, #65601) per plate for 2 h at 4 °C under rotation. For subsequent Western blot analysis immunoprecipitates were washed three times with lysis buffer and eluted from the beads in 1X NuPAGE LDS sample buffer (Thermo Fisher, NP0008) at 70 °C. For downstream proteomic experiments, beads were washed once in lysis buffer followed by two washing steps with 50 mM ammonium bicarbonate. Beads were resuspended in 150 μl of 50 mM ammonium bicarbonate, snap-frozen and stored at-80 °C. For on-bead digestion, 8 M urea/100 mM Tris-HCl pH8.2 was added to a final concentration of 2 M urea. Reduction and alkylation were carried out using 2 mM TCEP and 10 mM Chloroacetamide for 1 h at 30 °C under agitation in the dark. The solutions were diluted with Tris-HCl pH8.2 in a 1/1 ratio and digestion was performed with 1 μg trypsin per sample overnight at 30 °C under agitation in the dark. The next day, supernatant was taken from the beads and pooled with two washing steps with 100ul 10% ACN/Tris-HCl (final concentration of 3%ACN) and acidified to 0.5% TFA. Sample cleanup was performed using Sep-Pack C18 columns and completely dried using speed vac centrifugation. Samples were dissolved in LC-MS solution (3% ACN; 0.1% FA) for further analysis.

## Mass spectrometry

Dissolved samples were injected by an Easy-nLC 1000 system (Thermo Scientific) and separated on an EasySpray-column (75 μm × 500 mm) packed with C18 material (PepMap, C18, 100 Å, 2 μm, Thermo Scientific). The column was equilibrated with 100% solvent A (0.1% formic acid (FA) in water). Peptides were eluted using the following gradient of solvent B (0.1% FA in ACN): 5–25% B in, 60 min; 25–35% B in 10 min; 35–99% B in 5 min at a flow rate of 0.3 μl/min. All precursor signals were recorded in the Orbitrap using quadrupole transmission in the mass range of 300-1500 m/z. Spectra were recorded with a resolution of 120 000 at 200 m/z, a target value of 5E5 and the maximum cycle time was set to 3 s. Data dependent MS/MS were recorded in the linear ion trap using quadrupole isolation with a window of 1.6 Da and HCD fragmentation with 30% fragmentation energy. The ion trap was operated in rapid scan mode with a target value of 8E3 and a maximum injection time of 80 ms. Precursor signals were selected for fragmentation with a charge state from +2 to +7 and a signal intensity of at least 5E3. A dynamic exclusion list was used for 25 s. After data collection peak lists were generated using FCC[44] and Proteome Discoverer 2.1 (Thermo Scientific).

The raw-files from the mass spectrometer were converted into Mascot generic files (mgf) with Mascot Distiller software 2.4.2.0 (MatrixScience Ltd., London, UK). The peak lists were searched using Mascot Server 2.3 against the forward UniProtKB/Swiss-Prot database for human, concatenated to a reversed decoyed FASTA database (NCBI taxonomy ID 9606, release date 2012-04-12). The parameters for precursor tolerance and fragment ion tolerance were set to ± 10 ppm and ± 0.05 Da, respectively. Carbamidomethylation of cysteine was set as fixed modification, while oxidation (M) and Biotin (K) were set as variable. Enzyme specificity was set to trypsin, allowing up to 2 missed cleavages. The results were loaded into Scaffold 4.0 (Proteome Software, Portland, US) and filtered for peptide probability higher than

0.1% FDR, protein probability greater than 1% FDR and minimum of 1 peptide per protein. Relative quantification of fold change between sample groups based on total spectrum counts was performed using uncorrected t-test, using a significance threshold of p < 0.05, and assigning a minimum value of 0.1.

### Immunofluorescence

Cells grown in 4-well chamber slides (BD Falcon #08-774-25) were washed with PBS and fixed with 4% formalin for 15 min. Cells were then first incubated with 0.1 M glycine in PBS for 5 min and then permeabilized with 0.1% Triton-X100 in PBS for 10 min. After incubation with 4% horse serum in 0.1% Triton-X100 for 15 min, cells were incubated over night at 4 °C with primary antibody in 4% horse serum in a humid chamber. Primary antibodies used included anti-p300 (Santa Cruz; sc-8981; 1:250), anti-Flag M2 (Sigma F9291; 1:500) and anti-PAX3-FOXO1 (Clone PFM.2, CancerTools.org #160866; 1:500). The next day, cells were washed three times with PBS and incubated for 1 h with secondary antibody in 4% horse serum in the dark. Secondary antibodies used included Alexa568-labelled anti-rabbit IgG (ThermoFisher A-11011; 1:500) and Alexa594-labelled anti-mouse IgG (ThermoFisher A-11032; 1:500). The chamber was then removed and the slides were embedded with Vectashield with DAPI and imaged by epifluorescence microscopy (Nikon Eclipse Ti2) using the NIS-Elements AR 5.21.02 software.

For detection of P3F in transcriptional hubs, cells were grown in ibidi slides (ibidi 80827) coated with poly-L-lysine. After fixation and permeabilization as described above, samples were blocked with blocking buffer containing 1% BSA, 0.1% casein and 0.2% fish skin gelatin in PBS for 15 min. Afterwards, cells were incubated with anti-PAX3-FOXO1 antibody (CancerTools.org #160866; 1:500) in blocking buffer over night at 4 °C in a humid chamber. After washing and incubation with anti-mouse secondary antibody in blocking buffer as described above, cells were embedded with ibidi mounting medium (ibidi 50001) and imaged by super-resolution microscopy (Zeiss Elyra 7) using the Zeiss Zen black 3.5.SR software.

### Recruitment assay

For the recruitment assay, U2OS cells containing a 200 copy transgene array with a total of 50,000 repeats of the Lac operon integrated into chromosome 1 were used (kind gift of the laboratory of David L. Spector)[15]. 100,000 of these cells were plated per chamber of a 4-well chamber slide (BD Falcon #08-774-25) and 24 h later transfected with the different expression constructs using lipofectamine 3000 (ThermoFisher L3000-01). As bait proteins, CFP-LacI fusion proteins were assembled in a pSV2-based mammalian expression vector (kind gift of the laboratory of Richard Young) allowing visualization by fluorescence microscopy. To test for recruitment of endogenous p300 by the FOXO1 part of P3F, constructs driving the expression of different CFP-LacI-FOXO1 fusion proteins were transfected. After 48 h, cells were fixed for immunofluorescence with an anti-p300 antibody (Santa Cruz sc-8981;1:250) as described above and imaged by epifluorescence microscopy (Nikon Eclipse Ti2). For evaluation of recruitment of the different p300 domains by the FOXO1 part of P3F, cells were co-transfected with a construct driving the expression of the CFP-LacI-FOXO1 C-terminal AD fusion protein and a construct driving the expression of one of the different mCherry-NLS-p300 domain fusion proteins assembled in the plasmid mCherry2-C1 (Addgene #54563). After 48 h, cells were fixed and fluorescent proteins imaged as above.

### RNA-seq

Total RNAs were extracted using the RNeasy Plus Mini kit (Qiagen). For analysis of expression changes after shRNA-mediated PAX3-FOXO1 silencing or after knockouts of p300/CBP, two batches of paired-end polyA mRNA libraries were prepared using the NEBNext Ultra II Directional RNA Library Prep kit for Illumina. The libraries were sequenced on two flow cells of a Novaseq system as 2×150 base reads by GenomScan (Leiden, Netherlands). After the Illumina adaptors were trimmed off and quality control was performed with fastqc v0.11.7(https://www.bioinformatics.babraham.ac.uk/projects/fastqc/), the RNA-seq reads were aligned to the GRCh38 reference genome using STAR version 2.5.3a. Transcript per million (TPM) reads counts per gene were measured using RSEM version 1.3.3 by reassigning multiple alignments via a maximum likelihood estimation framework. After filtering for coding genes, we used edgeR[45] to estimate the dispersion and calculate the p-vales and FDR, filtering for differentially expressed genes with FDR < 0.05. For the double-knockout experiment of p300 and CBP, the RNA-seq reads were aligned to the GRCh38 reference genome using Hisat2 v2.1.0 with the options "--rna-strandness FR --fr"[46]. The mapped reads were sorted using Samtools v1.9[47]. Mapping quality was assessed by Qualimap v2.2.1[48]. Read counts were measured at the gene level using the Ensembl100 gene annotations and feature Counts v1.6.3 with the options "-s 1 -p -B -O -M --fraction"[49]. Read counts of each gene were combined from two technical replicates for each biological replicate for subsequent analysis. After genes with less than a sum of 10 reads from all libraries were filtered out, differential gene expression across samples was analyzed by the R package DESeq2 v3.7 using the variance stabilizing transformation method for normalization and taking the batch effect into account in the experimental design of the analysis[50]. Significant differential gene expression was defined by |fold-change| ≥ 1.5 and false discovery rate ≤ 0.05. RNA-seq for HAT inhibitors and degraders was analyzed by aligning reads to the GRCh38 reference genome using STAR version 2.5.3a. Transcript per million (TPM) reads counts per gene were measured using RSEM version 1.3.3. To compare gene set enrichment changes in two samples, we used GSEA (Gene Set Enrichment Analysis) tool. Visualization and summary of GSEA results were performed by using custom R scripts (https://github.com/GryderArt/VisualizeRNA-seq).

### AQuA-HiChIP

HiChIP experiments were performed using absolute quantification of chromatin architecture (AQuA)-HiChIP. Drugged human chromatin was spiked with chromatin extracted from C2C12 cells, a mouse muscle myoblast cell line, at a defined and conserved ratio of 1:10 mouse chromatin to human chromatin. Paired-end reads were mapped to the human genome (build hg38) and mouse genome (build mm10) to align human genomic content and mouse spike-in controls respectively, using Bowtie2 within the HiC-Pro pipeline[51]. RH4 and RH5 cells were treated with DMSO, dCBP1, or A485 and RH4-PAX3-FOXO1-FKBP12 cells were treated with DMSO or dTag-47. All treatments were for 6 h. Libraries were prepared using the Dovetail™ HiChIP MNase Kit User Guide (Version 1.2). Samples were incubated overnight with antibodies targeting RNA Pol2 (Santa Cruz, cat. #sc-47701 X; 9 µg antibody per 0.9 µg chromatin). HiChIP fastq files were aligned to the human genome version hg38 using BWA version 0.7.17 and were visualized in IGV. The peak densities were calculated by igvtools after fragment and shifting correction and were normalized with mouse spike-in following the previous study (PMID: 31784732). The significance of peaks was tested using MACS3 version 3.0.0a6, https://github.com/taoliu/MACS) with the -B option, a regular TF-binding mode for all targets reported in this paper. Pol2 peaks not captured by TF mode were considered by manual correction using the pooled peaks from the various experiments (we validate the performance using an alternative peak finder such as HOMER tools version 4.9.1 (http://homer.ucsd.edu/homer/index.html). Peaks of the spurious mapping artifacts were removed (reference locations "black-listed" by the ENCODE consortium https://sites.google.com/site/anshulkundaje/projects/blacklists). Motif analysis was performed using the HOMER tools. Mapped reads were then filtered for junction validity, and read pairs with contact range of 1000 bp or less were removed to filter out spurious MNase ligation

products. PCR duplicate-read pairs of punctate HiChIP pull-downs were not removed since they piled up at ChIP-seq peaks rather than acting as randomly distributed noise. Final read pairs (allValidPairs) were converted to Juicebox compatible.hic files for visual inspection using the hicpro2juicebox script provided within HiC-Pro. All downstream analyses requiring sub-matrix extraction from.hic files were performed using Juicer and straw[52]. Using peak3D pipeline Pol2 3D contacts were clustered. Briefly, 3D contacts above the preselected AQuA-CPM threshold were clustered when either of the ends were connected. The clusters having more than one contact edge were called 3D-clusters, and remaining edges were named as loops.

### ChIP-seq

Chromatin was extracted from adenine base edited RH4 cells with wild-type P3F or C793R mutant P3F or RH4 cells treated with DMSO or dCBP1 for 6 h. Samples were spiked with chromatin extracted from *Drosophila* cells at a defined and conserved ratio of 1:10 *Drosophila* chromatin to human chromatin. Paired-end reads were mapped to the human genome (build hg38) and *Drosophila* genome (dm3) to align human genomic content and *Drosophila* spike-in controls respectively, using BWA and were visualized in IGV. Samples were incubated overnight with antibodies targeting RNA Pol2 (Santa Cruz, cat. #sc-47701 X; 9 µg antibody per 0.9 µg chromatin) or H3K27ac (Millipore, cat #MABE647 RM172; 2.7 µg antibody per 2.7 µg chromatin). The peak densities were calculated by igvtools after fragment and shifting correction and were normalized with drosophila spike-in following the previous study (PMID: 31784732). The significance of peaks was tested using MACS3 version 3.0.0a6, https://github.com/taoliu/MACS) with the -B option, a regular TF-binding mode for all targets reported in this paper. Peaks not captured by TF mode were considered by manual correction using the pooled peaks from the various experiments (we validate the performance using an alternative peak finder such as HOMER tools version 4.9.1, http://homer.ucsd.edu/homer/index.html). Peaks of the spurious mapping artifacts were removed (reference locations "black-listed" by the ENCODE consortium https://sites.google.com/site/anshulkundaje/projects/blacklists).

### Heatmap of peaks

Centering on merged Pol2, p300, P3F and H3K27ac peaks or TSSR, we drew heatmaps of ChIP-seq data in different conditions using plotHeatmap.pl tools (https://github.com/GryderArt/ChIPseqPipe) implemented with a deeptools package (https://deeptools.readthedocs.io/en/develop/index.html). The program combines densities from the different bam files using bedCovComp pipeline (a wrapper program of the bedtools multicov tool), which reads multiple bam files to generate (spike-in) normalized densities at the peak regions.

### Quantification of Pol2 traveling and UnLoading ratios

Using hg38 RefSeq gene annotation, we split the gene coordinates into the promoter region ranging between −800 bp to −30 bp from the transcription start site (TSS), the TSS region (TSSR) between −30 bp and 300 bp from the TSS, the gene body between +300 bp and the transcription end site (TES), and the TES region (TESR) between TES and +4000 bp from the TES. Traveling ratio (TR) is defined as the ratio of Pol2 amount at TSSR and the amount of Pol2 in the remaining gene body. The Pol2 UnLoading ratio (PULR) is defined as the amount of Pol2 locating the promoter region, TSSR and over the amount of Pol2 in the TESR.

### Statistical analysis

Data analysis was performed with GraphPad Prism 8 (except for sequencing and mass spectrometry data). Significance was calculated using paired two-tailed Student $t$ test. Two-way ANOVA was used for multiple comparisons. Differences were considered statistically significant with $p < 0.05$. Data is represented as mean ± SD, unless otherwise noted in the figure legend.

### Reporting summary

Further information on research design is available in the Nature Portfolio Reporting Summary linked to this article.

## Data availability

The RNA-seq data generated in this study has been deposited in the European Nucleotide Archive (ENA) database with the accession number PRJEB47795. AQuA-HiChIP and ChIP-seq data is deposited with GEO NCBI under the accession number GSE208146. Source data are provided with this paper.

## Code availability

Code used throughout paper is available at https://github.com/GryderLab as noted in Methods above and at https://github.com/GryderLab/rms_additional_code.

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

## Acknowledgements

We thank Batool Akhtar-Zaidi for careful reading of the manuscript and helpful discussions and Santiago Ardiles for helpful tips for super-resolution microscopy. The work was supported by grants from the Swiss National Science Foundation (3100-175558 to B.W.S.), the Cancer League Switzerland (KLS-5143-08-2020 to B.W.S.), a clinical research focus grant from the University Zurich (to B.W.S) and the Childhood Research Foundation Switzerland (to B.W.S.). I.O. is supported as the Kasey Altman Scholar through Reign in Sarcoma. B.E.G. is supported through the DOD's Convergent Science Virtual Cancer Center, Reign in Sarcoma, the V Foundation and Alex's Lemonade Stand Foundation. PDX models generation at Institut Curie received support from MAP-PYACTS trial and H2020-lMI2-JTl-201 5-07 (116064-ITCC P4) grant.

## Author contributions

Conceptualization: K.B., B.W.S., B.E.G. and M.W. Methodology: I.O., K.B., D.L., Q.A.N., B.W.S., S.M., D.C., B.E.G. and M.W. Software: H.K., Y.A., Q.A.N., B.E.G., R.S., M.C., J.P., B.U. Formal analysis: H.K., Y.A., K.B., Q.A.N., D.L., J.W., M.W., S.M., B.E.G., R.S., J.P. Investigation: I.O., K.B., D.L., D.C., J.W., S.M., S.G.D., M.W. Resources: D.S., S.Z., O.D. Data curation: I.O., H.K., Y.A., Q.A.N., B.E.G., R.S., M.C., J.P., B.U. Writing-

original draft: I.O., K.B., Q.A.N., B.W.S., M.W., B.E.G. Writing-review and editing: I.O., K.B., Q.A.N., D.L., J.W., D.S., S.G.D., B.W.S., M.W., B.E.G., D.C., Y.A., R.S., J.P., M.I.H.K. Visualization: I.O., K.B., J.W., S.M., M.W., Y.A., B.E.G., R.S., M.C., J.P., H.K., S.M., B.U., M.I.H.K. Supervision: B.W.S., M.W., R.S., B.E.G. Project administration: B.W.S., R.S., B.E.G. Funding acquisition: B.W.S., B.E.G.

## Competing interests

The authors declare no competing interests.
