## [Peer Review File · Nature Communications]

PAX3-FOXO1 uses its activation domain to recruit CBP/P300 and shape RNA Pol2 cluster distributionREVIEWER COMMENTS

Reviewer #1 (Remarks to the Author):

The paper by Benischke and colleagues examines the molecular mechanisms by which the PAX3-FOXO1 fusion protein controls transcriptional activation of genes. This fusion protein is the product of a chromosomal translocation associated with a poor-prognosis paediatric soft tissue cancer called alveolar rhabdomyosarcoma. This fusion protein has a strong ability to reshape the genomic landscape of cells (e.g. PMID: 36395771, PMID: 35604932, PMID: 34386729, PMID: 28446439), as such the fate of many cells (e.g. PMID: 29869612). Thus, the study by Benischke and colleagues would not only have implications in the field of rhabdomyosarcoma but, more broadly, is of interest to researchers studying the transcriptional regulation of cell fate decision by transcription factors. Recent papers argue that this is an excellent "model system" for this issue.

In this paper, they further investigate the means by which PAX3-FOXO1 can induce transcriptional activity of cis-regulatory regions of the genome and activation of target gene transcription. The article is divided into two main parts. On the one hand, they perform a structure-function analysis highlighting the role of a single amino acid in the FOXO1 transactivation domain required for gene activation by PAX3-FOXO1 and maintenance of a particular cellular state and shape. On the other hand, they are studying the role of CBP/P300 in the transcriptional activity of PAX3-FOXO1 dependent cis-regulatory regions and highlighting in particular a role in the regulation of Pol2 genomic distribution.

The paper is well conducted and contains many interesting results:

- (i) a screening of the structural domains of PAX3-FOXO1 that are required for fitness/survival of RH4 cell lines (a Rhabdomyosarcoma cell line expressing PAX3-FOXO1)
- (ii) data showing that switching the cysteine C793 to a Serine (S) in PAX3-FOXO1 alters PAX3-FOXO1 activity.
 - a) PAX3-FOXO1C793S has a reduced ability compared to PAX3-FOXO1 to induce transactivation of the reporter transgene,
 - b) it cannot rescue several traits (transcriptome/cell shape) triggered by the loss of PAX3-FOXO1 in a rhabdomyosarcoma cell line (Rh4),
 - c) using an artificial context (yet interesting system) when the exact cysteine is mutated in a LacI-CFP-FOXO1 fusion protein the ability of LacI-CFP-FOXO1 protein recruit p300 is reduced
- (iii) data showing that inhibition of CBP and P300 activity alters the expression of PAX3-FOXO1, the expression of PAX3-FOXO1 target genes and the relocation of Pol2 from PAX3-FOXO1 targeted CRMs to CpG rich regions of the genome.

Revisions can be advised to clarify the statement, better connect the two parts and discuss their results more critically.

1) Where is PAX3-FOXO1C793S brought into the genome? Is it recruited to similar regions as the wild-type form of PAX3-FOXO1? How does its recruitment affect the activity of cis regulatory modules normally bound by PAX3-FOXO1? This could be assessed in the fibroblasts as in PMID: 28446439. How do P300, Pol2 partition in the presence of PAX3-FOXO1 C793S?.

Answers to these questions could bridge the two parts of the paper, and inform on the functional role of the C793 in the "natural" context.

2) CBP/P300 are central/core regulators of gene activation in cells. How expected are the results of the second part? Can the authors put more emphasis what's going on outside PAX3-FOXO1 target regions and discuss more this.

3) Regarding the "differentiation phenotype" of RH4 cells upon loss of PAX3-FOXO1 and gain for PAX3-FOXO1C793S (Notably figure2C), shP3F+ and shP3F+/ev conditions should be shown. Are the cells really related to muscle cells? Is the term differentiation adapted?

4) The discussion should not be just a restatement of the results. Authors could reconcile propose a model for the mode of action of PAX3-FOXO1 at the genome level.

Reviewer #2 (Remarks to the Author):

Manuscript: PAX3-FOXO1 uses its activation domain to recruit CBP/P300 and shape RNA Pol2 2 cluster distribution.

In this manuscript, Benischke, Osman et al. present their findings on the role of the PAX3-FOXO1 activation domain in the recruitment of CBP/P300 and its impact on the distribution of RNA Polymerase 2 clusters. The study highlights the importance of a specific cysteine residue within the AD domain of PAX3-FOXO1 that mediates P300 recruitment and transcriptional activation, ultimately leading to the maintenance of oncogenic transcriptional networks and cell survival. The figures presented in the paper are well-designed and effectively convey the research findings.

However, it should be noted that the AD-domain and C739 residue have previously been characterized in FOXO1 and linked to CBP-driven transcription, thereby impacting the novelty of this work. Additionally, there is a disconnect between the genomics work in the final three figures and the earlier figures that focus on the C739S residue in the AD-domain of P3F. The last three figures in the manuscript are of broader interest and transcend the biology of P3F-driven oncogenesis, and address potentially interesting biology around how P300/CBP regulate transcription by influencing the 3D genome around Pol2 occupied sites. In contrast the first figures of the manuscript lack novelty as the transactivation domain and critical residues herein had already been identified in the WT FOXO1 protein and the data suggests that the identified cysteine residue only moderately impacts P300/CBP recruitment and evaluation of its importance on a genome-wide level is lacking. The manuscript would benefit from more in-depth investigation of the genomic and transcriptomic consequences of P3F + CBP/P300 biology and the AD mutant herein, whereas the domain/residue identification could be condensed.

Figure 1: The domain-scanning approach is convincing and the fine-mapping of the important residues in the FOXO1 AD-domain is well done. It is important to control for localisation of the deletion mutants. Do these still localise to the nucleus and do they not aggregate, does deletion of this domain affect recruitment to transcriptional condensates? The importance of this domain is presumably independent from the oncogenic fusion studied here (as becomes apparent when studying C793). To this end, the authors could perform VP64 or wt-FOXO1 experiments and comment on the importance of the PAX3 DNA-binding domain in this experiment. Would the reporter be activated by a direct fusion of the AD to the PAX3 DNA-binding domain and do the inactive mutants act as dominant negatives?

Figure 2: The identified cysteine residue important in the AD domain of FOXO1 was previously demonstrated to be critical for wt-FOXO1 transcriptional activity. The impact of mutating this C612 residue was demonstrated to be dependent on CBP (CREBBP) but also contextual in nature as it can be influenced by the chromatin-context. How does the mutation of the C793 residue compare to a deletion of the region identified in the screen in the same assay? It appears that the mutant retains a substantial amount of transcriptional activity, yet it does not rescue differentiation induced by P3F depletion or proliferation, is the ASS1 region less sensitive? What happens to ASS1 mRNA expression in the complementation experiments? Also, can the authors comment on the fact that the rescue with P3F wt does not restore proliferation? Similar to figure 1 the authors should check the localisation of the mutants. Figure S3A-C should be in the main figures as it is the only endogenous experiment.

Figure 3: Due to the timing of the experiment it is difficult to separate direct from indirect targets. If available, the authors should look at directly bound P3F target genes (data from Figure 4F). The authors only use an FDR cutoff in their analysis, but an cutoff taking into consideration the magnitude of the effect would be important. Moreover, separation into most sensitive and least sensitive genes combined with an analysis of the contextual nature of the P3F binding sites may reveal whether certain genes are more or less affected by the C793 mutation and possibly why.

Figure 4: the BioID-MS experiment should be validated by western blot for P300 and also a target that is not affected by the mutation. It appears that the impact of the C793 mutant on p300 recruitment is relatively minor in the LacI-CFP assay as compared to deletion of the whole domain.

Does this mean that other residues are critical beyond C793? Is the recruitment of p300 and the importance of this residue dependent on the redox state in the cell? It has been previously demonstrated for other FOXO family members that cysteine residues may mediate protein interactions differentially based on the redox state (Dansen and colleagues). Is CBP also identified in the BioID-MS data and if so is it affected by the C793 mutation?

The authors demonstrate that p300 binding is enriched at sites occupied by P3F, how does this compare with acetylation levels and chromatin accessibility at these sites? Is there a difference between enhancers and promoter regions and importantly, how is p300 recruitment affected in a genome-wide manner by the C793 mutation as compared to WT?

Figure 5-6: The impact of P300i and P300-PROTACs on histone acetylation H3K18ac or H3K27ac should be assessed and segregated by enhancer and promoter regions. Recent papers should be discussed that demonstrate that acute transcriptional consequences are dependent on RNA Polymerase II initiation and pause-release, in part via BRD4 (Narita et al. and Hogg et al. Mol Cell).

The dCBP1 compound appears to impact the expression of P3F. Is this on a transcriptional or protein level? Could the observed effects be in part indirect? Does acute degradation of P3F via dTAG or Auxin systems result in similar genomic consequences as P300/CBP targeting? Would this depletion also impact 3D genome-organisation at Pol2 sites and similarly would catalytic inhibition of P300/CBP elicit the same effects?

In addition, it was recently demonstrated by Pelham-Webb et al. Mol Cell 2021 that A485 had minimal impact on the ability to re-establish the 3D genome organisation, this should be discussed in the context of results presented here. What happens to 3D genome looping at Pol2 sites strongly bound by P300 but that are not occupied by P3F. Are they affected in a similar manner?

Minor comments:

In figure 1: It appears that the authors tested only N-terminal deletion mutants, but the screen indicates that the N-terminal residues are not important for survival.

Line 213: Citation 11 appears to unrelated to the base-editing statement.

The authors should cite relevant recent literature on the targeting of P300/CBP and its impact on transcription via Pol2.

Reviewer #3 (Remarks to the Author):

The manuscript by Benischke et al. studies the role of the fusion transcription factor P3F, which is the result of a chromosomal translocation event between PAX3 and FOXO1 in the aggressive alveolar rhabdomyosarcoma (RMS). The paper highlights the role of a cysteine (C793) located at the C-terminus of P3F important in cell proliferation, gene regulation and interaction with the CBP/P300 coactivator. Inhibition of CBP/P300 compromised RMS cell proliferation which, according to the authors, would make it a putative target for therapeutic intervention. Mechanistically, CBP/P300 inhibition affects P3F-enriched Polymerase 2 clusters - essential for oncogenic transcription - promoting their collapse.

The study is well conducted through state-of-the-art methodology, the topic is relevant and provides new insights into the role of this fusion protein in RMS disease. However, some of the conclusions reached by the authors require additional experimental support, namely:

1- The authors conclude on page 16, bottom "Taken together, our data suggest that protein-protein interactions of P3F with CBP/p300 lead to acetylation of chromatin that facilitates Pol2 recruitment..." To confirm that P3F is recruiting P300 to its target genes and concomitantly increased histone acetylation, ChIP-seq of P300 and H3K27ac in the presence and absence of P3F in RH4 cells is required.

2-In Figure S2C-D, the authors show P3F depletion induced by dox in RH4 cells (blots on the right). Two bands are observed, one is labeled as P3F and the other as FOXO1, the latter showing a consistent increase upon P3F knockdown. The question remains whether this increase in FOXO1 could be responsible for at least part of the observed effects and attributed exclusively to P3F (either up or down P3F-dependent genes). The authors must address this point. Along the same lines, it should be clarified the antibody used in the ChIP-seq and/or the data analysis carried out, which allow the identification of PAX3-FOXO1 sites in RH4 cells shown in Figures 3F, 4F and 6F.

3- A new axis is proposed as a therapeutic tool that includes the use of inhibitor or degrader of CBP/P300 in RMS. This acetyltransferase is present in most of the enhancers, superenhancers and mediates the function of several nuclear receptors and participates in different cell functions. It is not clear to this reviewer how it could be used to treat alveolar RMS patients without expecting extensive effects. It is important to clarify this point, otherwise, this statement should be modified accordingly.

4-In the recruitment assays (Figure 4C), the total P300 signal comes from immunostaining, I wonder how it is almost exclusively detected at the insertion site of the 50,000 copies of the lac operon? The authors should clarify this point. In addition, CFP-FOXO1 constructs including the C793 mutant are used in Figure 4D. Shouldn't these experiments be performed with the P3F protein, to put the mutation in the right context? Since the mass spec showing the interaction between P3F with P300 were generated in HEK293 cells, unrelated system to the topic of the study, it is recommended to perform reciprocal coimmunoprecipitation (co-IPs) experiments in RH4 cells.

5- In Figure 5I, it is observed that P3F levels decreased in the presence of CBP inhibitor in RH4 cells. How does this result affect the conclusions that can be drawn throughout the study on the role assigned to CBP/P300? In addition, some of the blots in this Panel should be repeated since some bands are not correctly displayed (especially FOXO1 blot).

6- Does the presence of the CBP/P300 inhibitor affect P3F binding to its target sites in the genome? ChIPs of P3F in the presence and absence of dCBP1 at the target genes will help to clarify this point.

7- The models in Figure 7D-E are based on the results presented previously (Figure 6F, G, H and I and Figure 7 A-B), it is not clear to this reviewer that CBP/P300 inhibition-induced migration of the RNAPol 2 cluster from enhancers to promoter-enriched CpGs preferentially in P3F target genes, beyond the examples shown for SOX8 and MYOD1 (Figure 7A-B). Controls that support the proposed hypothesis are required.

Initial Response: We would like to express our gratitude to the reviewers and the senior editor of *Nature Communications* for their interest in our study as well as their very thorough and thoughtful comments. We have now addressed all raised concerns either by adopting the text or by providing novel experimental data. More specifically, our point-by-point responses to the reviewer comments are detailed below. In addition, we also added three additional authors to the author list, who all contributed to the revision experiments (Sarah Morice, Bhavatharini Udhayakumar, and Md Imdadul H. Khan).

REVIEWER COMMENTS

Reviewer #1 (Remarks to the Author):

The paper by Benischke and colleagues examines the molecular mechanisms by which the PAX3-FOXO1 fusion protein controls transcriptional activation of genes. This fusion protein is the product of a chromosomal translocation associated with a poor-prognosis paediatric soft tissue cancer called alveolar rhabdomyosarcoma. This fusion protein has a strong ability to reshape the genomic landscape of cells (e.g. PMID: 36395771, PMID: 35604932, PMID: 34386729, PMID: 28446439), as such the fate of many cells (e.g. PMID: 29869612). Thus, the study by Benischke and colleagues would not only have implications in the field of rhabdomyosarcoma but, more broadly, is of interest to researchers studying the transcriptional regulation of cell fate decision by transcription factors. Recent papers argue that this is an excellent "model system" for this issue.

In this paper, they further investigate the means by which PAX3-FOXO1 can induce transcriptional activity of cis-regulatory regions of the genome and activation of target gene transcription. The article is divided into two main parts. On the one hand, they perform a structure-function analysis highlighting the role of a single amino acid in the FOXO1 transactivation domain required for gene activation by PAX3-FOXO1 and maintenance of a particular cellular state and shape. On the other hand, they are studying the role of CBP/P300 in the transcriptional activity of PAX3-FOXO1 dependent cis-regulatory regions and highlighting in particular a role in the regulation of Pol2 genomic distribution.

The paper is well conducted and contains many interesting results:

- (i) a screening of the structural domains of PAX3-FOXO1 that are required for fitness/survival of RH4 cell lines (a Rhabdomyosarcoma cell line expressing PAX3-FOXO1)
- (ii) data showing that switching the cysteine C793 to a Serine (S) in PAX3-FOXO1 alters PAX3-FOXO1 activity.
 - a) PAX3-FOXO1C793S has a reduced ability compared to PAX3-FOXO1 to induce transactivation of the reporter transgene,
 - b) it cannot rescue several traits (transcriptome/cell shape) triggered by the loss of PAX3-FOXO1 in a rhabdomyosarcoma cell line (Rh4),
 - c) using an artificial context (yet interesting system) when the exact cysteine is mutated in a LacI-CFP-FOXO1 fusion protein the ability of LacI-CFP-FOXO1 protein recruit p300 is reduced
- (iii) data showing that inhibition of CBP and P300 activity alters the expression of PAX3-FOXO1, the expression of PAX3-FOXO1 target genes and the relocation of Pol2 from PAX3-FOXO1 targeted CRMs to CpG rich regions of the genome.

Revisions can be advised to clarify the statement, better connect the two parts and discuss their results more critically.

- 1) Where is PAX3-FOXO1C793S brought into the genome? Is it recruited to similar regions as the wild-type form of PAX3-FOXO1? How does its recruitment affect the activity of cis regulatory modules normally bound by PAX3-FOXO1? This could be assessed in the fibroblasts as in PMID: 28446439. How do P300, Pol2 partition in the presence of PAX3-FOXO1 C793S?.

Answers to these questions could bridge the two parts of the paper, and inform on the functional role of the C793 in the “natural” context.

-> We agree with the reviewer that these are interesting aspects. To address the functional role of C793 in an in vivo setting, we mutated C793 in RH4 cells by base editing and used the chromatin from these cells for ChIP-seq analysis. In particular, we determined genome-wide Pol2-partition as functional read-out downstream of P3F. These analyses revealed that Pol2 “piles up” at P3F binding sites and in parallel accumulates at transcriptional start sites near CpG islands suggesting that reduced p300 recruitment affects Pol2 pause release at these sites. The data from this novel experiment is shown in the novel Figures 3H (PIPOX gene tracks as example for a P3F regulated gene), Supplementary S5C (genome wide analysis at P3F binding sites) and 3I (genome wide analysis at TSS). We added the following text on page 8: “Then, we compared RNA Pol2 ChIP-seq between RH4 cells containing either wild-type or C793R mutant P3F. While we saw that the C793R mutation induced a small, genome-wide increase in Pol2 binding at P3F binding sites (Figures 3H and Supplementary S5C), the most striking finding was a substantial increase of Pol2 near TSS which contain overlapping CpG islands (Figure 3I).”

Since this latter effect is exactly what we also find upon p300 degradation by dCBP-1 (Figure 7C), we added this information on page 17: “Further analysis showed that, while Pol2-loss sites did contain cis-regulatory sequences (P3F motifs, MYOD and MYOG motifs, and SOX8 motifs) (Figure 7B) for the expected genes, Pol2-gained sites were at NFY motifs and frequently (83%) within CpG islands (Figure 7C), similar to the finding in cells with C793 mutant P3F (Figure 3I). The latter is consistent with a model in which Pol2 is strongly dependent upon acetylation for binding to non-CpG dense chromatin.”

2) CBP/P300 are central/core regulators of gene activation in cells. How expected are the results of the second part? Can the authors put more emphasis what’s going on outside PAX3-FOXO1 target regions and discuss more this.

To our knowledge, the interplay between P3F and p300 as well as the selective effect of p300 inhibitors on expression of genes regulated by P3F via Pol2 inactivation fits well into the common model for p300 mechanism of action. The redistribution of Pol2 to unmethylated CpG islands has, to our knowledge, not been described yet and is therefore surprising. The functional consequences of the redistribution is unclear.

As suggested by the reviewer, we discuss the effects outside of P3F target regions in more detail. We integrated this into a discussion of the similar and different effects on Pol2 distribution detected after p300/CBP degradation and C793 mutation on page 20-21: “Along these lines, our data also revealed a dramatic effect of the P3F-p300/CBP axis on Pol2 hubs. Pol2 clustering is an important determinant for the transcriptional output of genes. More than 80 molecules of Pol2 are estimated to condense in 300-400 nM droplets (Cho et al eLife 2016), forming liquid-like condensates at super-enhancers 19, and it has been established that the AD of TFs drives this (Bojja et al cell 2018). Our data showing the consequences of near complete degradation of p300 by the recently advanced CBP/p300 degrader dCBP1 17 confirms this mechanism and reveals a reduced recruitment of Pol2 to P3F binding sites. The more P3F sites overlapped Pol2 clusters, the more downregulated these clusters were after a 6-hour treatment with dCBP1. In contrast, the effects upon C793 mutation are more subtle. The mutation does not affect recruitment of Pol2 to P3F sites, but rather leads to its trapping at these sites, suggesting that under conditions of reduced but not completely absent p300, Pol2 pause release is more affected than recruitment. This data highlights the importance of CBP/p300 for Pol2 clustering at P3F target genes in FP-RMS and also confirms the previous finding that the ability of enhancers to activate genes-at-a-distance depends on CBP/p300 (Karr et al Genes & Development 2022).

After both p300 degradation and C793 mutation we also detected a striking accumulation of Pol2 at CpG islands of CpG island-containing promoters of genes within Pol2 transcriptional clusters, which is accompanied with a collapse of long-range Pol2 contacts. These genes, despite abundant Pol2 in the promoter and gene body, lack binding of Pol2 past the transcriptional end site, and potentially fail to

produce mature RNA transcripts. RNA Pol2's retreat to CpG islands upon CBP/p300 degradation is likely a result of these islands remaining accessible despite a lack of acetylation on nearby histones. The loss of acetylation collapsing Pol2 into CpG islands on chromatin is in stark contrast to the effect of hyper-acetylation on Pol2 that removes Pol2 from both enhancers and promoters of these same genes (Gryder et al. *Nature Genetics* 2019), indicating that the dynamic opposition of HATs and HDACs at Pol2 clusters is striking a critical balance of acetylation (Garzón-Porrás et al. *ACS Chemical Biology* 2022). These data paint a mechanistic picture of a fusion-oncogene as an enabler of 3D Pol2 hubs via long-range cis-recruitment of histone acetylation enzymes. This highlights the fusion oncoprotein P3F as a prominent guiding force for the strength of Pol2 clustering, and for the interaction frequency between enhancer-bound Pol2 and promoter-bound Pol2 in FP-RMS cells."

3) Regarding the 'differentiation phenotype' of RH4 cells upon loss of PAX3-FOXO1 and gain for PAX3-FOXO1C793S (Notably figure 2C), shP3F+ and shP3F+/ev conditions should be shown. Are the cells really related to muscle cells? Is the term differentiation adapted?

As requested by the reviewer, we added pictures for shP3F+ and shP3F+/ev conditions to create the new figure 2C. The legend was adapted accordingly.

Rhabdomyosarcoma cells indeed are related to muscle cells, as has been recognized for a long time and recently been characterized in more detail in a series of single cell analysis papers (Patel et al, *Dev Cell* 2022, Wei et al, *Nat Cancer* 2022, Danielli et al. *Sci Adv* 2023, DeMartino et al *Nat Comm* 2023). All these studies demonstrated that RMS tumors are composed of different subpopulations reminiscent of different stages along the myogenic lineage, including stem-cell like cells, cycling and non-cycling committed progenitor cells, and differentiated cells. The latter are characterized by muscle differentiation markers like myosin-light and -heavy chains, which are therefore commonly used to evaluate the extent of "differentiation" in RMS. Different treatments including inhibition of the ras pathway (both for aRMS and eRMS) as well as interference with PAX3-FOXO1 in case of aRMS leads to induction of differentiation-related genes. Nevertheless, RMS cells are not bona fide muscle cells and there might also be relevant differences, such as cell fusion and multinucleation, which are not frequently seen in RMS. Overall however, the term differentiation is accepted in the field and we believe its use is adequate here.

4) The discussion should not be just a restatement of the results. Authors could reconcile propose a model for the mode of action of PAX3-FOXO1 at the genome level.

We re-structured the discussion and integrated different additional aspects (also raised by the other reviewers).

Concerning the mode of action of P3F, which we already tried to depict in Fig. 8E, we added the following paragraph: "Overall, our data characterizes CBP/p300 as an important amplifier of P3F mediated gene expression and demonstrates that recruitment of p300/CBP to its target genes is a very important function of P3F. This complements existing data of the mode of action of P3F in RMS cells. Chromatin binding of P3F has been shown to be affected by the chromatin remodeler CHD4 (Marques JG et al. *eLife* 2020). The following recruitment of p300/CBP to these sites then promotes PIC assembly, leading to Pol2 superclustering. BRD4 activity at these sites is also required, but for RNA Pol2 cluster flexibility that enables transcription of CR TFs in RMS, and BRD4 degradation results in a failure of Pol2 to unload past the TESR of CR TFs (Chin DH et al. *Pharmaceuticals* 2023). Interference with any of these steps affects P3F driven gene expression and therefore might have therapeutic promise for patients with FP-RMS."

Reviewer #2 (Remarks to the Author):

Manuscript: PAX3-FOXO1 uses its activation domain to recruit CBP/P300 and shape RNA Pol2 2 cluster distribution.

In this manuscript, Benischke, Osman et al. present their findings on the role of the PAX3-FOXO1 activation domain in the recruitment of CBP/P300 and its impact on the distribution of RNA Polymerase 2 clusters. The study highlights the importance of a specific cysteine residue within the AD domain of PAX3-FOXO1 that mediates P300 recruitment and transcriptional activation, ultimately leading to the maintenance of oncogenic transcriptional networks and cell survival. The figures presented in the paper are well-designed and effectively convey the research findings.

However, it should be noted that the AD-domain and C739 residue have previously been characterized in FOXO1 and linked to CBP-driven transcription, thereby impacting the novelty of this work. Additionally, there is a disconnect between the genomics work in the final three figures and the earlier figures that focus on the C739S residue in the AD-domain of P3F. The last three figures in the manuscript are of broader interest and transcend the biology of P3F-driven oncogenesis, and address potentially interesting biology around how P300/CBP regulate transcription by influencing the 3D genome around Pol2 occupied sites. In contrast the first figures of the manuscript lack novelty as the transactivation domain and critical residues herein had already been identified in the WT FOXO1 protein and the data suggests that the identified cysteine residue only moderately impacts P300/CBP recruitment and evaluation of its importance on a genome-wide level is lacking. The manuscript would benefit from more in-depth investigation of the genomic and transcriptomic consequences of P3F + CBP/P300 biology and the AD mutant herein, whereas the domain/residue identification could be condensed.

Figure 1: The domain-scanning approach is convincing and the fine-mapping of the important residues in the FOXO1 AD-domain is well done. It is important to control for localisation of the deletion mutants. Do these still localise to the nucleus and do they not aggregate, does deletion of this domain affect recruitment to transcriptional condensates?

-> We performed immunofluorescence detection of all individual deletion mutants in 293T cells. This approach revealed that full-length P3F as well as most deletion mutants are predominantly, but not exclusively, localized in the nucleus. Exceptions from this pattern are found in case of P3F delta 667 AA, which is predominantly localized in the cytoplasm, demonstrating that the nuclear localization signal is affected in this mutant. Furthermore, the newly generated P3-AD 60AA fusion protein (used for reporter assays as described below) is exclusively localized in the nucleus, suggesting that the large disordered part of FOXO1 influences subcellular localization to some extent in 293T cells. Finally, the AD 138AA domain is diffusely expressed. None of the deletion mutants aggregated under these conditions. Pictures from the immunostainings are shown in novel supplementary Figure S1F.

We added the following sentence to the text on page 6: "Importantly, with exception of P3F-Δ667AA all the deletion mutants are predominantly (from full-length P3F down to P3F- Δ445AA) or partially (AD 138AA) localized in the nucleus (Supplementary Figure S1F)."

Since the ectopically expressed deletion mutants are highly overexpressed, detection of nuclear condensates was challenging. As an alternative, and likely even more relevant, we studied nuclear condensates of endogenous P3F in aRMS cells and compared the wildtype with the C793 mutant form. For this we used a recently described, PAX3-FOXO1 specific antibody recognizing a breakpoint-spanning epitope (Azorsa et al, Modern Pathology 2021) to detect the fusion protein by immunofluorescence. The staining nicely reveals the nuclear condensates as speckles in the nucleus. We used super-resolution microscopy to evaluate the staining and found that similar to wildtype P3F

also the C793 mutant form is still localized in transcriptional condensates. Microscopic pictures are shown in the novel supplementary Figure S3B.

We added the following sentence to the text on page 7: "Expression of P3F protein remained intact and detectable in the nucleus as transcriptional hubs as determined by super-resolution microscopic analysis of immunofluorescence staining using a P3F-specific antibody (Azorsa et al, Modern Pathology 2021) (Supplementary Figure S3A-B)."

The importance of this domain is presumably independent from the oncogenic fusion studied here (as becomes apparent when studying C793). To this end, the authors could perform VP64 or wt-FOXO1 experiments and comment on the importance of the PAX3 DNA-binding domain in this experiment. Would the reporter be activated by a direct fusion of the AD to the PAX3 DNA-binding domain and do the inactive mutants act as dominant negatives?

-> We thank the reviewer for these important questions and performed the following experiments: First, to formally prove that wildtype FOXO1 is not binding to the PAX3-responsive promoter used for our reporter assay, we tested the transactivation potency of FOXO1 (containing mutations of three important AKT-sites regulating its subcellular localization (T24A, S256A and S319A) to induce nuclear localization, commonly used for such assays) with the ASS1-reporter construct. Our data clearly shows that FOXO1 transactivates this reporter only at very low levels, demonstrating that the PAX3 DNA binding domain is necessary.

Next, we tested the transactivation potency of the AD alone fused directly to the PAX3 DNA-binding domains of the fusion protein. Interestingly, compared to full-length P3F, this short fusion protein has a similar activity, suggesting that in this assay the disordered part of FOXO1 does not contribute to its activity. The data from these two novel experiments are shown in the novel Figure 1E and a scheme depicting the structure of the P3-AD fusion was added to Figure 1C.

We added the following sentence to the text on page 6: "However, when fused to the PAX3 part of P3F, a 60AA long fragment spanning the AD had a transactivation potency similar to full-length P3F (Fig. 1E), confirming that this small part suffices for induction of transactivation and demonstrating that the whole disordered part of FOXO1 is not relevant, at least in this assay. Wildtype FOXO1 did not transactivate this reporter, highlighting the importance of specific DNA binding (Figure 1E)."

Finally, we added different amounts of the inactive P3F- Δ 80AA deletion mutant to full-length P3F to evaluate whether the inactive mutants act as dominant negatives. Indeed, we detected a P3F- Δ 80AA dose-dependent decrease of transactivation, suggesting that this mutant interferes with full-length P3F. The relatively mild effect suggests that this is mainly via competition for binding to the promoter in the reporter construct. The data from this experiment is shown in the novel Figure 1F.

We added the following sentence to the text on page 6: "Furthermore, addition of different amounts of the inactive P3F delta 80AA deletion mutant to full-length P3F reduced transactivation in a dose-dependent manner (Figure 1F), as expected in case of competition for DNA binding."

Figure 2: The identified cysteine residue important in the AD domain of FOXO1 was previously demonstrated to be critical for wt-FOXO1 transcriptional activity. The impact of mutating this C612 residue was demonstrated to be dependent on CBP (CREBBP) but also contextual in nature as it can be influenced by the chromatin-context. How does the mutation of the C793 residue compare to a deletion of the region identified in the screen in the same assay?

-> To compare the effect of C793 mutation with indel mutants generated in the domain screen, we mutated C793 in Rh4 cells using the base editing system and tested the effect of the mutation on proliferation/survival of the cells by a competition assay similar to the one used in the screen. This approach revealed an about 8-fold depletion of cells containing the mutation-inducing sgRNA compared to control cells. In the original indel-based screen the drop-out rate was about 30-fold for sgRNAs located

in the AD domain. However, in such an indel-based domain screen 2/3 indels result in a frameshift, which generally increases the depletion rate, also for sgRNAs targeting non-structured (non-essential) regions. This is reflected by depletion rates of 5-10 for sgRNAs located in the disordered region of P3F (see Fig.1B). Taking this into account, we concluded that the physiological effects of indel-based disturbance of the AD and base-editing mediated mutation of C793 are comparable. The novel data is shown in the novel supplementary Figure S3C.

We added the following sentence to the text on page 7: "In a competition assay similar to the one used for the screen to evaluate effects on cell survival, mutation of C793 led to an 8-fold depletion of affected cells compared to control cells, confirming the relevance of intact C793 in an endogenous context (Supplementary Figure S3C)."

It appears that the mutant retains a substantial amount of transcriptional activity, yet it does not rescue differentiation induced by P3F depletion or proliferation, is the ASS1 region less sensitive? What happens to ASS1 mRNA expression in the complementation experiments?

->In the RNAseq data ASS1 levels are only slightly affected by P3F silencing. When measured by qRT-PCR, effects are much stronger. The reason could be that P3F binds to intron 11 of ASS1 and induces expression of a transcript containing only the downstream exons, while the expression of the full length transcript is unaffected. Nevertheless, ASS1 transcription is Cys793 sensitive and not rescued by the Cys mutant P3F. We generated an additional figure depicting the behavior of different individual genes upon silencing/rescue of P3F, including ASS1. The data is shown as supplementary figure S5B.

Also, can the authors comment on the fact that the rescue with P3F wt does not restore proliferation? We can only speculate here, however many oncogenes are only tolerated in a certain range of expression. Both too high and too low levels are not well tolerated (described in detail for EWS-FLI1 and called there "goldilocks principle"; see Bo Kyung A. Seong et al. Cancer Cell 2021). The most probable explanation is that we are outside of the well-tolerated dosage range.

Similar to figure 1 the authors should check the localisation of the mutants.

We agree with the reviewer that the aspect of subcellular localization is important. Since most of the tested mutations do not affect PAX3-FOXO1 activity, we focused on C793 mutant PAX3-FOXO1. We mutated C793 by base editing and as stated above, used the PAX3-FOXO1 specific antibody to detect the fusion protein by immunofluorescence and super-resolution microscopy. This approach demonstrated that C793 mutation does not lead to a redistribution of the fusion protein, which remains in the nucleus. Immunofluorescence images of wildtype and C793 mutant PAX3-FOXO1 as well as Sanger sequencing reads to confirm the C793 mutation in this experiment are shown in the novel supplementary Figure S3B.

We added the following sentence to the text on page 7 (as described already above): "Expression of P3F protein remained intact and detectable in the nucleus as transcriptional hubs as determined by super-resolution microscopic analysis of immunofluorescence stainings using a P3F-specific antibody (Azorsa et al, Modern Pathology 2021) (Supplementary Figure S3A-B)."

Figure S3A-C should be in the main figures as it is the only endogenous experiment.

As suggested by the reviewer, we moved Figures S3A-C to Figure 2 as new panels G-I.

Figure 3: Due to the timing of the experiment it is difficult to separate direct from indirect targets. If available, the authors should look at directly bound P3F target genes (data from Figure 4F). Using P3F binding sites from prior data, we classified failed and partial rescue genes as direct targets if they had a P3F binding site within 500kb of their gene TSS. We provided an annotation of this as a column in Supplementary Table 2.

The authors only use an FDR cutoff in their analysis, but a cutoff taking into consideration the magnitude of the effect would be important. Moreover, separation into most sensitive and least sensitive genes combined with an analysis of the contextual nature of the P3F binding sites may reveal whether certain genes are more or less affected by the C793 mutation and possibly why. While we thank the reviewers for their suggestion, our motif analysis did not yield any significant difference between motifs at peaks near direct vs. indirect genes and was therefore omitted from the manuscript.

Figure 4: the BioID-MS experiment should be validated by western blot for p300 and also a target that is not affected by the mutation.

-> As requested by the reviewer, we validated the BioID-MS experiment by Western blot. The validation data is in agreement with the MS data and confirms that wt P3F-BirA biotinylates p300 much stronger than C793S mutant P3F-BirA. In contrast, auto-biotinylation of P3F is unaffected by C793S mutation. As additional, not affected control, we used HDAC2, a component of the NuRD complex. The NuRD complex has been shown to functionally interplay with P3F, albeit not via direct protein-protein interaction, but via binding to DNA in close proximity to P3F sites (Böhm M et al. JCI 2016). Importantly, biotinylation of HDAC2 is much less affected by the C793 mutation. All this novel data is shown in the novel blot in Fig.4B.

We added the following sentence to the text on page 11: "BioID followed by Western blot confirmed the reduced ability of C793 mutant P3F-BirA to biotinylate p300 compared to the wildtype P3F-BirA. In contrast, auto-biotinylation was unaffected, while biotinylation of HDAC2, which as a member of the NuRD complex is detected at genomic sites in close proximity to P3F binding sites (Böhm M et al. JCI 2016, and Marques JG et al. eLife 2020), was much less affected by the mutation compared to p300 (Figure 4B)."

It appears that the impact of the C793 mutant on p300 recruitment is relatively minor in the LacI-CFP assay as compared to deletion of the whole domain. Does this mean that other residues are critical beyond C793? Is the recruitment of p300 and the importance of this residue dependent on the redox state in the cell? It has been previously demonstrated for other FOXO family members that cysteine residues may mediate protein interactions differentially based on the redox state (Dansen and colleagues). Is CBP also identified in the BioID-MS data and if so is it affected by the C793 mutation?

-> Indeed, there is some activity remaining in C793 mutant P3F, not only in the ability to recruit p300, but also in transactivation activity, as shown in Fig. 2B. We believe that the helical structure of the AD is of importance and that absence of C793 affects this structure to some extent (as shown in Fig.2F by structural prediction), but does not lead to complete unfolding. It cannot be excluded that additional residues affect this structure.

To evaluate the influence of the redox state on P3F activity, we tested the effect of N-acetylcysteine (NAC) on transcriptional activity of P3F in the reporter assay. A similar assay is presented in the paper by Dansen et al, mentioned by the reviewer. There, NAC decreased the inhibitory effect of p300 on FOXO4 as determined by a FOXO4-responsive reporter. In contrast to these results, we did not detect any influence of NAC on P3F in our assay (see figure below). We therefore concluded that the redox state has no strong effect on P3F activity and decided to omit this aspect from the manuscript.

Figure legend: Reporter assay performed with indicated amounts of plasmid coding for full-length P3F in 293T cells. Transfected cells were cultured in presence or absence of 1.25 mM N-acetylcysteine (NAC) (n=2).

CBP is indeed also detected by BioID-MS, however there was no significant difference between wildtype and mutant P3F in number of spectral counts for the peptides specifically assigned to CBP (see below).

Figure legend: Normalized total spectrum counts for p300 and CBP as measured in BioID-MS experiments in 293T cells. BioID was performed with wt and C793 mutant P3F-BirA and biotinylated proteins were quantified by MS (n=3).

Due to the high homology between p300 and CBP, different shorter peptides detected by MS could originate from both CBP or p300. Still, the number of specific peptides for each protein (23 in case of CBP, 35 in case of p300) is large enough for a clear conclusion. Importantly however, as shown in supplemental Fig.S6, only double knock-out of p300 and CBP in Rh4 cells affects expression P3F target genes, strongly suggesting that in aRMS cells both proteins have similar co-factor functions for P3F (both are involved under basal conditions) or alternatively, are able to replace each other (p300 is preferred, but CBP replaces it when absent).

The authors demonstrate that p300 binding is enriched at sites occupied by P3F, how does this compare with acetylation levels and chromatin accessibility at these sites?

-> To address this question, we compared the H3K27ac levels as well as chromatin accessibilities as assessed by ATAC-seq between p300-bound sites with and without P3F. The analysis revealed a large difference in acetylation levels, while in contrast there is no difference in chromatin accessibility. This is in agreement with data shown in other publications, which overall show that only few genomic regions experience loss of chromatin accessibility after p300/CBP inhibition (Martire S et al., BMC Molecular and Cell Biology 2020 and Narita T et al., Mol Cell 2021). Hence, p300 recruitment by P3F promotes transcription, but not chromatin opening, which probably is regulated at a more upstream level, maybe at some extent even before P3F binds to the DNA, e.g. by the chromatin remodeler CHD4 (see Marques JG et al, eLife 2020). This novel data was added to Fig 4F.

We added the following sentence to the text on page 11: "Strikingly, p300 levels are 2-3 times higher at P3F sites than at control sites, in accordance with higher H3K27Ac levels, while no difference in DNA accessibility was detected (Figure 4F)."

Is there a difference between enhancers and promoter regions and importantly, how is p300 recruitment affected in a genome-wide manner by the C793 mutation as compared to WT?

-> To address the functional role of C793 in an in vivo setting, we mutated C793 in Rh4 cells by base editing and used the chromatin from these cells for ChIP-seq analysis. Instead of p300, we focused on Pol2-partition as functional consequence. These analyses revealed that Pol2 "piles up" at P3F binding sites and in parallel accumulates at transcriptional start sites near CpG islands suggesting that reduced p300 recruitment affects Pol2 pause release at these sites. The data from this novel experiment is shown in the novel Figures 3H (PIPOX gene tracks as example for a P3F regulated gene), Supplementary S5C (genome wide analysis at P3F binding sites) and 3I (genome wide analysis at TSS). We added the following text on page 8: "Then, we compared RNA Pol2 ChIP-seq between RH4 cells containing either wild-type or C793R mutant P3F. While we saw that the C793R mutation induced a small, genome-wide increase in Pol2 binding at P3F binding sites (Figures 3H and Supplementary S5C), the most striking finding was a substantial increase of Pol2 near TSS which contain overlapping CpG islands (Figure 3I)."

Since this latter effect is exactly what we also find upon p300 degradation by dCBP-1 (Figure 7C), we added this information on page 17: "Further analysis showed that, while Pol2-loss sites did contain cis-regulatory sequences (P3F motifs, MYOD and MYOG motifs, and SOX8 motifs) (Figure 7B) for the expected genes, Pol2-gained sites were at NFY motifs and frequently (83%) within CpG islands (Figure 7C), similar to the finding in cells with C793 mutant P3F (Figure 3I). The latter is consistent with a model in which Pol2 is strongly dependent upon acetylation for binding to non-CpG dense chromatin."

Figure 5-6: The impact of P300i and P300-PROTACs on histone acetylation H3K18ac or H3K27ac should be assessed and segregated by enhancer and promoter regions.

-> We performed the requested analysis with H3K27Ac levels in RH5 cells (which, aside from a MYCN amplification, are phenotypically similar to RH4 cells). The analysis shows that p300 degradation reduces H3K27ac levels to a similar extent at both promoter-proximal and -distal sites. This novel data is shown in the novel Fig.6H. We added the following sentence to the text on page 15: "Comparing Pol2 HiChIP to H3K27ac HiChIP, we observed that following dCBP1 treatment H3K27ac contact signal is overwhelmingly lost whereas RNA Pol2 contact signal appears to increase and decrease in near equal measure across sites (Figure 6H)."

Recent papers should be discussed that demonstrate that acute transcriptional consequences are dependent on RNA Polymerase II initiation and pause-release, in part via BRD4 (Narita et al. and Hogg et al. Mol Cell).

-> We would like to thank the reviewer for these insightful hints. We discussed the data of the two mentioned papers in the context of our findings and added to following paragraphs to the discussion on page 19: "Apart from p300, our BioID experiment identified other potential interactors of P3F that depend on C793, including MED6, a component of the mediator complex and TAF9, a component of TFIID, that

supports the assembly of the pre-initiation complex (PIC). Recent studies demonstrated that p300/CBP, besides acetylation-dependent BRD4 recruitment, also promotes assembly of the PIC by acetylation of sites subsequently bound by the bromodomain protein TAF1, another component of TFIID (Narita et al Mol Cell 2021). Accordingly, acute transcriptional consequences of decreased histone acetylation levels at promoter regions after p300 inhibition are due to both loss of Pol2 binding and reduction of pause release mediated by BRD4 (Narita et al Mol Cell 2021 and Hogg et al Mol Cell 2021). This suggests that MED6 and TAF9 follow p300 recruitment to P3F sites, reflecting the ongoing establishment of the transcriptional machinery. Along these lines, our data also revealed a dramatic effect of the P3F-p300/CBP axis on Pol2 hubs.”

The dCBP1 compound appears to impact the expression of P3F. Is this on a transcriptional or protein level?

-> Our data suggests that dCBP-1 affects P3F expression post-transcriptionally, since its mRNA is not much affected (see Fig.5J).

Could the observed effects be in part indirect?

-> The reviewer is right, the effects could be partially indirect, via reduced levels of P3F and/or MYCN. However, the effect on P3F is evident after 24h of treatment with dCBP-1 (Fig.5I), while after 6h, when molecular analyses were done, P3F levels are only very slightly affected. We realized that in the text the mentioned time point for Fig 5I (6 hours) was wrong, while in the legend it was correct (24 hours). We corrected this mistake and added a Western Blot performed with lysates from Rh4 cells after 6h of treatment with dCBP-1 to illustrate the difference (Supplementary Figure S7E).

Does acute degradation of P3F via dTAG or Auxin systems result in similar genomic consequences as P300/CBP targeting? Would this depletion also impact 3D genome-organisation at Pol2 sites and similarly would catalytic inhibition of P300/CBP elicit the same effects?

-> We used recently published, engineered Rh4 cells which contain a knock-in of FKBP12 into the P3F locus, allowing efficient degradation of P3F upon incubation with dTAG-47 small molecule (Zhang S et al. Mol Cell 2022). With these cells we performed HiChIP with an anti-Pol2 antibody after acute P3F degradation to determine effects on 3D genome organization and saw a similar collapse of Pol2 contacts as after p300 degradation. The novel data is shown in the novel Figure 8D. We added the following sentence on page 18:” Additionally, we performed Pol2 HiChIP on an RH4 cell line where P3F was engineered (Zhang et al Mol Cell 2022) to encode a protein tag FKBP12(F36V) after treating it with the degrader dTAG-47 (Supplementary Figure 10D) and saw a similar collapse of Pol2 signal at long-range contacts (25 Kb to 3Mb in distance, Figure 8D).”

In addition, it was recently demonstrated by Pelham-Webb et al. Mol Cell 2021 that A485 had minimal impact on the ability to re-establish the 3D genome organisation, this should be discussed in the context of results presented here.

-> We combined the aspect of 3D genome organization with DNA accessibility as opposed to transcriptional effects. We added to following paragraph to the discussion: “Interestingly, we did not detect an influence of p300/CBP on DNA accessibility, which is in agreement with previous studies showing that loss of chromatin accessibility after p300/CBP inhibition is small (Narita T et al., Mol Cell 2021 and Martire S et al, BMC Molecular and Cell Biology 2020). Furthermore, also 3D chromatin organization as measured by Hi-C has been shown to be largely independent of p300 activity (Pelham-Webb B et al., Mol Cell 2021). Taken together this suggests that chromatin opening is regulated at a more upstream level, before the recruitment of p300/CBP to regulatory sites.”

What happens to 3D genome looping at Pol2 sites strongly bound by P300 but that are not occupied by P3F. Are they affected in a similar manner?

-> We compared the effect of p300 degradation on 3D genome looping at Pol2 sites between p300 sites with and without P3F. This analysis revealed that the strongest effect is seen at P3F sites, while p300 only sites are slightly less affected. The data is shown as part of novel Fig. 7D.

We added the following sentence to the text on the page 17: "... regions which experienced the most loss of contacts among Pol2 clusters were those which contained both p300 and P3F binding at the ends of their loops (Figure 7D)"

Minor comments:

In figure 1: It appears that the authors tested only N-terminal deletion mutants, but the screen indicates that the N-terminal residues are not important for survival.

-> No, actually all deletions are C-terminal deletion mutants.

Line 213: Citation 11 appears to unrelated to the base-editing statement.

-> The citation was indeed wrong. We replaced it with the correct one: Huang, T.P., G.A. Newby, and D.R. Liu, Precision genome editing using cytosine and adenine base editors in mammalian cells. *Nat Protoc*, 2021. 16(2): p. 1089-1128.

The authors should cite relevant recent literature on the targeting of P300/CBP and its impact on transcription via Pol2.

-> We included citations describing the effect of p300 inhibition on transcription (Narita T et al., *Mol Cell* 2021).

Reviewer #3 (Remarks to the Author):

The manuscript by Benischke et al. studies the role of the fusion transcription factor P3F, which is the result of a chromosomal translocation event between PAX3 and FOXO1 in the aggressive alveolar rhabdomyosarcoma (RMS). The paper highlights the role of a cysteine (C793) located at the C-terminus of P3F important in cell proliferation, gene regulation and interaction with the CBP/P300 coactivator. Inhibition of CBP/P300 compromised RMS cell proliferation which, according to the authors, would make it a putative target for therapeutic intervention. Mechanistically, CBP/P300 inhibition affects P3F-enriched Polymerase 2 clusters - essential for oncogenic transcription - promoting their collapse.

The study is well conducted through state-of-the-art methodology, the topic is relevant and provides new insights into the role of this fusion protein in RMS disease. However, some of the conclusions reached by the authors require additional experimental support, namely:

1- The authors conclude on page 16, bottom "Taken together, our data suggest that protein-protein interactions of P3F with CBP/p300 lead to acetylation of chromatin that facilitates Pol2 recruitment..." To confirm that P3F is recruiting P300 to its target genes and concomitantly increased histone acetylation, ChIP-seq of P300 and H3K27ac in the presence and absence of P3F in RH4 cells is required.

→ We have now added support to this claim by performing ChIP-seq and HiChIP before and after mutation of P3F (C793S). The results are on display in both main and supplementary data figures throughout, and show the following: (1) at sites of p300 that lack P3F binding, we observe open chromatin but not substantial Pol2 binding nor H3K27ac marking; (2) P3F, when mutated to P3F C793R, causes a small increase in Pol2 positioning at P3F binding sites and a much larger increase at TSS with CpG islands (passively open chromatin), very similar to the effect of p300 degradation by dCBP-1 (3) degradation of P3F leads to a loss of Pol2 contacts at cluster sites. These together suggest that p300

recruitment by P3F is driving activity and disruption of canonical P3F activity through mutation or degradation collapses Pol2 away from typically acetylated chromatin and towards CpG islands. These data are provided in novel Figure 3I, Figure 4F, Supplementary Figure 5C, and Figure 8D.

2-In Figure S2C-D, the authors show P3F depletion induced by dox in RH4 cells (blots on the right). Two bands are observed, one is labeled as P3F and the other as FOXO1, the latter showing a consistent increase upon P3F knockdown. The question remains whether this increase in FOXO1 could be responsible for at least part of the observed effects and attributed exclusively to P3F (either up or down PF3-dependent genes). The authors must address this point.

-> We agree that is an interesting question. A similar upregulation of FOXO1 is detected upon mutation of C793 of PAX3-FOXO1 (Figure S4C). To address this aspect, we generated a FOXO1 knock-out variant of Rh4 cells by mutation of the splice site region of exon 1 in FOXO1 using the base editing system. The resulting cells do not express FOXO1, while P3F levels are unchanged compared to parental Rh4 cells. The data characterizing these cells is shown in the new supplementary Figures S4A-B. We then mutated C793 of P3F in these cells. Interestingly, muscle differentiation markers including MYH3, MYL1 and TNNC2, are about 10-fold less induced under these conditions compared to parental Rh4 cells. In contrast, there is only a small difference in the reduction of the mRNA level of direct P3F target genes. This data is also shown in the novel supplementary Figure S4C. The data suggests that FOXO1 plays an important role in the physiological effect downstream of P3F inactivation. Importantly, when using base editing directed towards C793 of P3F, the corresponding C612 in FOXO1 is co-mutated, suggesting that the activating function of FOXO1 is not relevant for these effects. Importantly, however, beside the transcriptional activation domain at its C-terminus, FOXO1 also acts as transcriptional repressor by recruiting co-repressors like Sin3A. Involved domains are located at the N-terminus. The net effect seems to be context dependent.

We added the following paragraph to the text on page 7: "Interestingly, both silencing and mutation of P3F lead to an upregulation of wildtype FOXO1 expression (Supplementary Figure S2C and 3A). To evaluate whether FOXO1 is involved in some of the downstream effects after P3F inactivation, we generated a FOXO1 knock-out variant of RH4 cells by mutation of the splice site of exon 1 using base editing (Supplementary Figure S4A-B). Interestingly, upregulation of muscle differentiation markers was strongly reduced in these cells upon mutation of C793 of P3F compared to parental cells, while effects on P3F target genes were less affected (Supplementary Figure S4C). This suggests that part of the physiological effect downstream of inactivation of P3F is indeed due to FOXO1 upregulation."

Along the same lines, it should be clarified the antibody used in the ChIP-seq and/or the data analysis carried out, which allow the identification of PAX3-FOXO1 sites in RH4 cells shown in Figures 3F, 4F and 6F.

-> For ChIPseq analysis of P3F binding sites a monoclonal P3F-specific Ab (PFM.2) was used. This Ab was first described for this purpose in Cao et al, Cancer Res 2010. The Ab has also been further characterized in Azorsa et al, Modern Pathology 2021 and is nowadays commercially available. The data used here has been generated in Gryder BE et al. Nat Genetics 2019.

3- A new axis is proposed as a therapeutic tool that includes the use of inhibitor or degrader of CBP/p300 in RMS. This acetyltransferase is present in most of the enhancers, superenhancers and mediates the function of several nuclear receptors and participates in different cell functions. It is not clear to this reviewer how it could be used to treat alveolar RMS patients without expecting extensive effects. It is important to clarify this point, otherwise, this statement should be modified accordingly.

-> We agree with the reviewer that this question is very important here. We included a more detailed discussion of this aspect in the discussion part. First, there are p300 inhibitors (targeting the bromodomain) in clinical trials, which are well tolerated. Furthermore, inhibition of p300/CBP has been shown to affect mainly expression of cell type specific genes (See Narita T et al., Mol Cell 2021). Also our data shows that in aRMS cells there is a preference for P3F target genes upon interference with

p300. Therefore, there might be a therapeutic window available that allows to target aRMS cells with a certain specificity. However, we did not detect cell death upon p300 inhibition, suggesting that from a therapeutic perspective a more specific interference with the P3F-p300 interaction is of higher interest. Importantly, an inhibitor interfering with the interaction of wildtype FOXO1 with p300 has recently been described (Jang JY et al. J.Clin.Invest 2022).

We changed the corresponding paragraph in the discussion to the following text: “The broad functionality of p300/CBP might speak against these proteins as direct targets for such a therapy. Importantly however, two p300/CBP bromodomain inhibitors CCS1477 and FT-7051 are currently in clinical trials for cancer patients, and preliminary data showed good tolerance (Armstrong et al AACR. Mol Cancer Ther. 2021; 20(12)Suppl):Abstract nr P202). Furthermore, we could show that CBP/p300 is bound to an even greater degree at P3F sites compared to non-P3F bound enhancers. Accordingly, CBP/p300 degradation caused efficient halting of P3F target genes and the FP-RMS core regulatory circuitry, while in comparison downregulation of housekeeping genes was only limited. This is in line with recent data showing that p300/CBP inhibition selectively downregulates cell type-specific genes (Narita T et al Mol Cell 2021) and suggests that there might be a therapeutic window available for p300/CBP inhibitors. Nevertheless, since we detected only cytostatic effects in FP-RMS cells, p300/CBP-inhibitors might not represent the first choice. Instead, compounds that interfere with the recruitment of p300 by the FOXO1 part of the fusion and mimicking genetic inactivation of the AD of P3F, might be more promising. Interestingly, a compound with such a mechanism of action has recently been described for wildtype FOXO1 (Jang JY et al. J Clin Invest 2022). Potentially, C793 and its surrounding domain could also be a target for an inhibitory compound. Whether C793 is reactive enough to serve as target for a covalent inhibitor similar to inhibitors developed for other oncogenes (Boike et al., Cell Chemical Biology, 2021) represents an interesting question for future studies.”

4-In the recruitment assays (Figure 4C), the total P300 signal comes from immunostaining, I wonder how it is almost exclusively detected at the insertion site of the 50,000 copies of the lac operon? The authors should clarify this point.

-> In general, p300 is concentrated at the lac operon insertion site, detected as a bright spot by immunofluorescence, while in the rest of the cells p300 is more diffusely distributed and less prominently detected. However, we agree with the reviewer that it's astonishing that in some cases most of the p300 is soaked up by FOXO1 like a sponge. However, it has to be taken into account that the FOXO1 fusion protein here is ectopically overexpressed and that the more than 50,000 lac binding sites in these cells represent a number that is probably not magnitudes away from the total p300 binding sites present in the genome. Furthermore, the number of FOXO1 proteins there might be further increased by partition into the locally formed phase separated compartment. Hence, it can be expected that a larger fraction of p300 moves to the lac operon insertion site.

In addition, CFP-FOXO1 constructs including the C793 mutant are used in Figure 4D. Shouldn't these experiments be performed with the P3F protein, to put the mutation in the right context?

-> No, P3F itself binds to the genome via the PAX3 DNA binding domains and this will disturb the read-out (single spot in the nucleus). This is therefore not possible.

Since the mass spec showing the interaction between P3F with P300 were generated in HEK293 cells, unrelated system to the topic of the study, it is recommended to perform reciprocal coimmunoprecipitation (co-IPs) experiments in RH4 cells.

-> Indeed, we tried several times to co-immunoprecipitate P3F and p300, but always failed. This is actually the reason why we selected the recruitment assay as an alternative method to validate the BioID results and study the interplay of the two proteins. While we do not completely exclude that under the right (mild enough) conditions co-immunoprecipitation is possible, we also concluded from our results that the P3F-p300 interaction is not stable enough to be maintained during cells lysis with commonly used lysis buffers containing (only mild) detergents. Co-immunoprecipitation might be in general not an

optimal method to study protein-protein interactions that take place in transcriptional hubs, since these are formed by phase separated compartments having (often) a fluid-like behavior and involve weak protein-protein interactions. Our experience from previous proteomic work (Böhm M et al. J. Clin Invest 2016) is that proteins co-bound to DNA in close proximity are predominantly co-immunoprecipitated with P3F. These interactions however are lost upon digestion of the DNA.

To clarify this point, we included the following sentence on page 11: “Since we were not able to confirm the interaction between P3F and p300 by co-immunoprecipitation, we performed a previously described recruitment assay to validate this interaction.”

5- In Figure 5I, it is observed that P3F levels decreased in the presence of CBP inhibitor in RH4 cells. How does this result affect the conclusions that can be drawn throughout the study on the role assigned to CBP/P300? In addition, some of the blots in this Panel should be repeated since some bands are not correctly displayed (especially FOXO1 blot).

-> We realized that in the text the mentioned time point for Fig 5I (6 hours) was wrong, while in the legend it was correct (24 hours). We agree that at the 24 hour time point (which was used as reference for the physiological experiments), in Rh4 and IC-pPDX-35 cells P3F levels itself are also affected by dCBP-1, while in the two other models there is no effect on P3F. Therefore, part of the long-term effects in the first two models could be the result of this downregulation of P3F. This finding however does not affect the conclusions about the relevance and mechanism of action of the AD including the relevant C793. Furthermore, most of the molecular analyses were made after 6h of treatment. In order to clarify the difference to the long term experiments, we added a Western Blot for Rh4 cells treated for 6 hours with dCBP-1, which demonstrates that at this time point the effect on P3F protein level is still very small, in contrast to the effects on p300 and NMYC. This is shown in the novel Supplementary Figure S7E.

6- Does the presence of the CBP/P300 inhibitor affect P3F binding to its target sites in the genome? ChIPs of P3F in the presence and absence of dCBP1 at the target genes will help to clarify this point.

-> In Pol2 binding data at P3F wt target sites observed between wild-type and C793R mutant P3F, we observed a small, genome-wide increase in Pol2 binding at P3F binding sites following the mutation, while the more striking finding was a substantial increase of Pol2 near TSS which contain overlapping CpG islands. This latter effect is exactly what we also find upon p300 degradation by dCBP-1 (Figure 7C). This suggests that while DNA binding activity of P3F is likely unaltered, Pol2 function is affected, potentially at the pause-release step.

7- The models in Figure 7D-E are based on the results presented previously (Figure 6F, G, H and I and Figure 7 A-B), it is not clear to this reviewer that CBP/P300 inhibition-induced migration of the RNApol 2 cluster from enhancers to promoter-enriched CpGs preferentially in P3F target genes, beyond the examples shown for SOX8 and MYOD1 (Figure 7A-B). Controls that support the proposed hypothesis are required.

-> We compared both the decrease of Pol2 and the extent of Pol2 accumulation in CpG island after p300 degradation at p300/P3F co-bound sites with p300 only sites and sites neither bound by p300 nor P3F. This analysis demonstrated that the decrease of Pol2 is highest at P3F target sites, while accumulation of Pol2 is highest at sites neither bound by p300 nor P3F. The net effect is movement of Pol2 away from P3F target genes to CpG island elsewhere in the genome. This novel data is shown in the novel Fig.7D.

REVIEWERS' COMMENTS

Reviewer #1 (Remarks to the Author):

The article by Benischke and colleagues has been considerably improved. All the reviewers' comments have been taken into account, from technical validations to reconciling the literature on transcriptional regulation mediated by PAX3-FOXO1, CBP/p300 and Pol2 in the context of a discussion. In particular, the article is now very coherent, from the identification of interactions between P3F and CBP/p300 to the role of this interaction in chromatin acetylation and Pol2 recruitment and clustering. The article provides new information on the implication of PAX3-FOXO1 fusion protein in Rhabdomyosarcoma cell state, but more generally on how transcription factors acting at superenhancers control Pol2 activity via its distribution in the genome. It is extremely relevant to the field of cancer epigenetics and genome regulation.

Reviewer #2 (Remarks to the Author):

The authors have convincingly answered my questions and concerns and should be commended for an excellent job during the review process.

Reviewer #3 (Remarks to the Author):

The manuscript authors have effectively addressed all of my previous comments and concerns by conducting additional experiments, as illustrated in the new Figures 3I, 4F, S5C, 8D, S4C, S7E, and 7D, and by providing clarifications in the text. In light of these improvements, I believe that the paper is now suitable for publication in Nature Communications.

Reviewer #1 (Remarks to the Author):

The article by Benischke and colleagues has been considerably improved. All the reviewers' comments have been taken into account, from technical validations to reconciling the literature on transcriptional regulation mediated by PAX3-FOXO1, CBP/p300 and Pol2 in the context of a discussion. In particular, the article is now very coherent, from the identification of interactions between P3F and CBP/p300 to the role of this interaction in chromatin acetylation and Pol2 recruitment and clustering. The article provides new information on the implication of PAX3-FOXO1 fusion protein in Rhabdomyosarcoma cell state, but more generally on how transcription factors acting at superenhancers control Pol2 activity via its distribution in the genome. It is extremely relevant to the field of cancer epigenetics and genome regulation.

Reviewer #2 (Remarks to the Author):

The authors have convincingly answered my questions and concerns and should be commended for an excellent job during the review process.

Reviewer #3 (Remarks to the Author):

The manuscript authors have effectively addressed all of my previous comments and concerns by conducting additional experiments, as illustrated in the new Figures 3I, 4F, S5C, 8D, S4C, S7E, and 7D, and by providing clarifications in the text. In light of these improvements, I believe that the paper is now suitable for publication in Nature Communications.

Response to reviewers:

We would like to thank all the reviewers for their hard work and their insightful advice on our manuscript. We believe that during the review process the manuscript improved considerably and are very happy with the outcome.